# DORA: Exploring Outlier Representations in Deep Neural Networks

**Kirill Bykov**                                    *kbykov@atb-potsdam.de*
*Understandable Machine Intelligence Lab*
*Leibniz Institute for Agriculture and Bioeconomy (ATB), Potsdam, Germany*
*Technical University of Berlin, Berlin, Germany*

**Mayukh Deb**                                    *mayukhmainak2000@gmail.com*
*Independent*

**Dennis Grinwald**                                    *dennis.grinwald@tu-berlin.de*
*Machine Learning Group*
*Technical University of Berlin, Berlin, Germany*
*BIFOLD – Berlin Institute for the Foundations of Learning and Data, Berlin, Germany*

**Klaus-Robert Müller**                                    *klaus-robert.mueller@tu-berlin.de*
*Machine Learning Group*
*Technical University of Berlin, Berlin, Germany*
*BIFOLD – Berlin Institute for the Foundations of Learning and Data, Berlin, Germany*
*Department of Artificial Intelligence, Korea University, Seoul 136-713, Korea*
*Max Planck Institut für Informatik, 66123 Saarbrücken, Germany*
*Google Research, Brain Team, Berlin, Germany*

**Marina M.-C. Höhne**                                    *mhoehne@atb-potsdam.de*
*Understandable Machine Intelligence Lab*
*Leibniz Institute for Agriculture and Bioeconomy (ATB), Potsdam, Germany*
*Department of Computer Science, University of Potsdam*
*Department of Physics and Technology, UiT Arctic University of Norway*
*BIFOLD – Berlin Institute for the Foundations of Learning and Data, Berlin, Germany*

**Reviewed on OpenReview:** *https://openreview.net/forum?id=nfYwRIezvg*

## Abstract

Deep Neural Networks (DNNs) excel at learning complex abstractions within their internal representations. However, the concepts they learn remain opaque, a problem that becomes particularly acute when models unintentionally learn spurious correlations. In this work, we present *DORA* (Data-agnOstic Representation Analysis), the first data-agnostic framework for analyzing the representational space of DNNs. Central to our framework is the proposed *Extreme-Activation* (EA) distance measure, which assesses similarities between representations by analyzing their activation patterns on data points that cause the highest level of activation. As spurious correlations often manifest in features of data that are anomalous to the desired task, such as watermarks or artifacts, we demonstrate that internal representations capable of detecting such artifactual concepts can be found by analyzing relationships within neural representations. We validate the EA metric quantitatively, demonstrating its effectiveness both in controlled scenarios and real-world applications. Finally, we provide practical examples from popular Computer Vision models to illustrate that representations identified as outliers using the EA metric often correspond to undesired and spurious concepts.

# 1 Introduction

The ability of Deep Neural Networks (DNNs) to perform complex tasks and achieve *state-of-the-art* performance in various fields can be attributed to the rich and hierarchical representations that they learn Bengio et al. (2013). Far beyond the handcrafted features that were inductively imposed by humans on learning machines in classical Machine Learning methods Marr and Nishihara (1978); Jackson; Fogel and Sagi (1989), Deep Learning approaches exploit the network's freedom to learn complex abstractions. However, a prevalent concern remains that the nature of concepts, learned by the model remains unknown. The rapid progress in representation learning only exacerbates the issue of interpretability, since DNNs are frequently trained using self-supervised methodologies Jaiswal et al. (2020); LeCun and Misra (2021) on immense volumes of data Brown et al. (2020); Bommasani et al. (2021), which accelerated the unpredictability concerning the scope of possible learned concepts and their mutual relations Goh et al. (2021).

The increasing popularity of Deep Learning techniques across various fields, coupled with the difficulty of interpreting the decision-making processes of complex models, has led to the emergence of the field of Explainable AI (XAI) (e.g. Montavon et al. (2018); Samek et al. (2019); Xu et al. (2019); Gade et al. (2019); Rudin (2019); Samek et al. (2021)). Research within XAI has revealed that the internal representations that form the basis of DNNs are susceptible to learning harmful and undesired concepts, such as biases Guidotti et al. (2018); Jiang and Nachum (2020), Clever Hans (CH) effects Lapuschkin et al. (2019), and backdoors Anders et al. (2022). These malicious concepts often are unnatural or anomalous in relation to the relevant concepts within the dataset. Examples include watermarks in the PASCAL 2007 image classification task Lapuschkin et al. (2019), Chinese logographic watermarks in ImageNet dataset Li et al. (2022), colored band-aids in skin-cancer detection problem Anders et al. (2022) or tokens in a pneumonia detection problem Zech et al. (2018).

To enhance our understanding of the decision-making processes within complex machines and to prevent biased or potentially harmful decisions, it is crucial to explain the concepts learned during training. By analyzing the relationships between internal neural representations, we can gain insights into the model's predictive strategies. In this work, we introduce *Representation Analysis*, a framework dedicated to exploring the representations of a particular model layer. Our approach utilizes a proposed *Extreme-Activation* (EA) metric, which measures the similarity between various learned representations within the networks by examining the common activation patterns on Activation-Maximisation Signals (AMS). These signals represent data points where the representations exhibit their highest activations and can be identified through either a *data-aware* (natural) process from an existing data corpus Borowski et al. (2020) or a *data-agnostic* process, in which the signals are synthetically generated Erhan et al. (2009); Olah et al. (2017). We refer to the representation analysis conducted with the latter method as *DORA** (Data-agnOstic Representation Analysis). We demonstrate the interpretability of our proposed distance measure and study the connections between natural and synthetic Activation-Maximisation Signals. Moreover, we quantitatively assess the alignment between the functional EA distance and human perception — we demonstrate that EA distances between representations generally align with human judgment regarding the similarity of concepts, particularly in scenarios where the concepts underlying the representations are known. Additionally, we highlight our proposed distance measure's ability to establish a robust baseline for detecting inserted anomalous concepts in controlled scenarios. Lastly, through practical experiments conducted on popular Computer Vision models, we reveal that anomalous representations identified by our framework often correspond to undesirable spurious concepts.

# 2 Related Work

To address the concerns regarding the black-box nature of complex learning machines Baehrens et al. (2010); Vidovic et al. (2015); Buhrmester et al. (2019); Samek et al. (2021), the field of *Explainable AI (XAI)* has emerged. While some recent research focuses on inducing the self-explaining capabilities through changes in the architecture and the learning process Gautam et al. (2022a;b); Chen et al. (2018); Gautam et al. (2021), the majority of XAI methods (typically referred to as *post-hoc* explanation methods) are decoupled from the

---

*PyTorch implementation of the proposed method can be found by the following link: `https://github.com/lapalap/dora` .

training procedure. A dichotomy of post-hoc explanation methods could be performed based on the scope of their explanations, i.e., the model behavior can be either explained on a *local* level, where the decision-making strategy of a system is explained for one particular input sample, or on a *global* level, where the aim is to explain the prediction strategy learned by the machine across the population and investigate the purpose of its individual components in a universal fashion detached from single datapoints (similar to feature selection Guyon and Elisseeff (2003)).

*Local* explanation methods typically interpret the prediction by attributing relevance scores to the features of the input signal, highlighting the influential characteristics that affected the prediction the most. Various methods, such as Layer-wise Relevance Propagation (LRP) Bach et al. (2015), GradCAM Selvaraju et al. (2019), Occlusion Zeiler and Fergus (2014), MFI Vidovic et al. (2016), Integrated Gradient Sundararajan et al. (2017), have proven effective in explaining Graph Neural Networks Wang et al. (2021); Tiddi et al. (2020) as well as Bayesian Neural Networks Bykov et al. (2021); Brown and Talbert (2022). To further boost the quality of interpretations, several enhancing techniques were introduced, such as SmoothGrad Smilkov et al. (2017); Omeiza et al. (2019), NoiseGrad and FusionGrad Bykov et al. (2022). Considerable attention also has been paid to analyzing and evaluating the quality of local explanation methods (e.g. Samek et al. (2016); Hedström et al. (2022); Guidotti (2021); Binder et al. (2023)). However, while the local explanation paradigm is incredibly powerful in explaining the decision-making strategies for a particular data sample, the main limitation of such methods is their inability to effectively investigate the unexplored behaviors of the models, such as the detection of previously unknown spurious correlations and computational shortcuts Adebayo et al. (2022).

*Global* explanation methods aim to interpret the general behavior of learning machines by investigating the role of particular components, such as neurons, channels, or output logits, which we refer to as representations. Existing methods mainly aim to connect internal representations to human understandable concepts, making the purpose and semantics of particular network sub-function transparent to humans. Methods such as Network Dissection Bau et al. (2017; 2018) and Compositional Explanations of Neurons Mu and Andreas (2020) aim to associate representations with human-understandable concepts. They achieve this by examining the intersection between the concept-relevant information provided by a binary mask and the activation map of the corresponding representation. The MILAN method Hernandez et al. (2021) generates a text description of the representation by searching for a text string that maximizes the mutual information with the image regions in which the neuron is active.

## 2.1 Activation-Maximisation Methods

The family of Activation-Maximization (AM) Erhan et al. (2009) methods aims to globally explain the concepts behind neurons by identifying the input that triggers maximal activation in a particular neuron or network layer, thereby visualizing the features learned. These inputs, which we will refer to as Activation-Maximization Signals (AMS), could be either natural, found in a *data-aware* fashion by selecting a "real" example from an existing data corpus Borowski et al. (2020), or artificial, found in a *data-agnostic* mode by generating a synthetic input through optimization Erhan et al. (2009); Olah et al. (2017); Szegedy et al. (2013).

In comparison to earlier synthetic AM methods, Feature Visualization (FV) Olah et al. (2017) performs optimization in the frequency domain by parametrizing the image with frequencies obtained from the Fourier transformation. This reduces adversarial noise in resulting explanations (e.g. Erhan et al. (2009); Szegedy et al. (2013)) — improving the interpretability of the obtained signals. Additionally, the FV method applies multiple stochastic image transformations, such as jittering, rotating, or scaling, before each optimization step, as well as frequency penalization, which either explicitly penalizes the variance between neighboring pixels or applies bilateral filters on the input.

## 2.2 Spurious Correlations

Deep Neural Networks are prone to learn spurious representations — patterns that are correlated with a target class on the training data but not inherently relevant to the learning problem Izmailov et al. (2022). Reliance on spurious features prevents the model from generalizing, which subsequently leads to poor performance on

sub-groups of the data where the spurious correlation is absent (cf. Lapuschkin et al. (2016; 2019); Geirhos et al. (2020)). In the field of Computer Vision, such behavior could be characterized by the model's reliance on aspects such as an images background Xiao et al. (2020), object textures Geirhos et al. (2018), or the presence of semantic artifacts in the training data Wallis and Buvat (2022); Lapuschkin et al. (2019); Geirhos et al. (2020); Anders et al. (2022). Artifacts can be added to the training data on purpose as Backdoor attacks Gu et al. (2017); Tran et al. (2018), or emerge naturally and might persist unnoticed in the training corpus, resulting in *Clever Hans effects* Lapuschkin et al. (2019).

Recently, XAI methods have demonstrated their potential in revealing the underlying mechanisms of predictions made by models, particularly in the presence of artifacts such as Clever Hans or Backdoor artifacts. Spectral Relevance analysis (SpRAy) aims to provide a global explanation of the model by analyzing local explanations across the dataset and clustering them for manual inspection Lapuschkin et al. (2019). While successful in certain cases Schramowski et al. (2020), SpRAy requires a substantial amount of human supervision and may not detect artifacts that do not exhibit consistent shape and position in the original images. SpRAY-based Class Artifact Compensation Anders et al. (2022) method allowed for less human supervision and demonstrated its capability to suppress the artifactual behavior of DNNs.

### 2.3 Comparison of Representations

The study of representation similarity in DNN architectures is a topic of active research. Numerous methods comparing network representations have been applied to different architectures, including Neural Networks of varying width and depth Nguyen et al. (2020), Bayesian Neural Networks Grinwald et al. (2022), and Transformer Neural Networks Raghu et al. (2021). Some works Ramsay et al. (1984); Laakso (2000); Kornblith et al. (2019); Nguyen et al. (2022) argue that the representation similarity should be based on the correlation of a distance measure applied to layer activations on training data. Other works Raghu et al. (2017); Morcos et al. (2018) compute similarity values by applying variants of Canonical Correlation Analysis (CCA) Hardoon et al. (2005); Bießmann et al. (2010) on the activations or by calculating mutual information Li et al. (2015), or employ kernel methods to quantify the evolution of the representations Montavon et al. (2011); Braun et al. (2008). However, these methods are predominantly utilized for the comparison of whole representation spaces, e.g. layers, and not individual components, and as a result often overlook the semantics of learned concepts. Furthermore, those methods are dependent on the availability of data.

## 3 Distance Metrics between Neural Representations

In the following, we start with the definition of a *neural representation* as a sub-function of a given network that depicts the computation graph, from the input of the model to the output of a specific neuron.

**Definition 1** (Neural representation). *We define a neural representation $f$ as a real-valued function $f : \mathbb{D} \to \mathbb{R}$, mapping from the data domain $\mathbb{D}$ to the real numbers $\mathbb{R}$.*

The following definition is introduced to highlight the distinction between the traditional notion of a neuron and the broader computational process encapsulated in the term *neural representation*. Conventionally, a neuron is defined as a function that takes inputs from its preceding neurons. In contrast, a neural representation describes the entire computational process, starting from the input and yielding the activation of a specific neuron (unit). While some neurons in DNNs produce multidimensional outputs, depending on the specific use cases, multidimensional functions could be regarded either as a set of individual representations or alternatively could be aggregated to achieve scalar output. For example, in the case of convolutional neurons that output activation maps containing the dot product between filter weights and input data at each location, activation maps could be aggregated by average- or max-pool operations for the sake of simplifying the explanation of the semantic concept underlying the function. The choice depends on the particular aim and scope of the analysis and does not alter the network itself.

The scalar output of representations often corresponds to the amount of evidence or similarity between concepts present in the input and internally learned abstractions. Various sub-functions within the model could be considered as neural representations, ranging from the neurons in the initial layers that are often regarded as elementary edge or color detectors Le and Kayal (2021), to the model output. Throughout this

work, we primarily focus on the high-level abstractions that emerge in the latest layers of networks, such as the feature-extractor layers in well-known Computer Vision architectures, as they are frequently employed for transfer learning Zhuang et al. (2020).

In DNNs, neural representations are combined into layers — collections of individual neural representations that typically share the same computational architecture and learn abstractions of similar complexity. In the scope of the following work, we mainly focused on the analysis of the relations between representations within one selected layer from the network.

**Definition 2** (Layer). *We define a layer $\mathcal{F} = \{f_1, ..., f_k\}$ as a set comprising $k$ individual neural representations.*

To examine the relationships between representations, we can begin by analyzing the behavior of functions with respect to a given dataset. We define a dataset, $D$, consisting of $N$ data points denoted as $D = \{x_1, ..., x_N\}$. This set is referred to as the *evaluation dataset* and is used to measure the relationship between two neural representations. We make the assumption that these data points, $x_1, ..., x_N$, are independently and identically distributed (i.i.d.) samples from the overall data distribution $\mathcal{D}$. Additionally, we standardize the activations of representations on this evaluation dataset, resulting in a mean of 0 and a standard deviation of 1.

For a neural representation $f_i \in \mathcal{F}$ and an evaluation dataset $D = \{x_1, ..., x_N\}$, we define a vector of activations

$$\mathbf{a}_i = (f_i(x_1), ..., f_i(x_N)),\tag{1}$$

where

$$\mu_i := \frac{1}{N}\sum_{t=1}^{N} f_i(x_t) = 0, \quad \sigma_i := \sqrt{\frac{1}{N-1}\sum_{t=1}^{N}(f_i(x_t) - \mu_i)^2} = 1.\tag{2}$$

Standardizing the vectors in this way can help to mitigate any differences in scale between the vector components and ensure that each component contributes equally to the distance calculation. Below, we present three widely recognized metrics that can be utilized to measure the distance between neural representations.

- **Minkowski distance**:

$$d_M(f_i, f_j) = \left(\sum_{t=1}^{N}|f_i(x_t) - f_j(x_t)|^p\right)^{\frac{1}{p}},\tag{3}$$

  where $p \geq 1, p \in \mathbb{Z}$ determines the degree of the norm, which gauges the sensitivity of the metric to differences between the components of the vectors being compared. In general, larger values of $p$ lead to a greater emphasis on larger differences between the components of the vectors. Conversely, smaller values of p reduce the influence of larger differences, leveraging an increased uniform weighting of all components.

- **Pearson distance**:

$$d_P(f_i, f_j) = \frac{1}{\sqrt{2}}\sqrt{1 - \rho_p(\mathbf{a}_i, \mathbf{a}_j)},\tag{4}$$

  where $\rho_p(\mathbf{a}, \mathbf{b})$ is the Pearson correlation coefficient between the vectors $\mathbf{a}$ and $\mathbf{b}$. The Pearson correlation coefficient is a widely used metric for measuring the linear dependence between two random variables. It is an interpretable measure of similarity, however, it is also sensitive to outliers, which can significantly affect the calculated distance.

- **Spearman distance**:

$$d_S(f_i, f_j) = \frac{1}{\sqrt{2}}\sqrt{1 - \rho_s(\mathbf{a}_i, \mathbf{a}_j)},\tag{5}$$

  where $\rho_s(\mathbf{a}, \mathbf{b})$ is the Spearman rank-correlation coefficient between vectors $\mathbf{a}$ and $\mathbf{b}$. The Spearman correlation is a non-parametric rank-based metric commonly used to measure the monotonic dependence between two random variables. Its main advantage is that it is robust to outliers and can handle ties in the data.

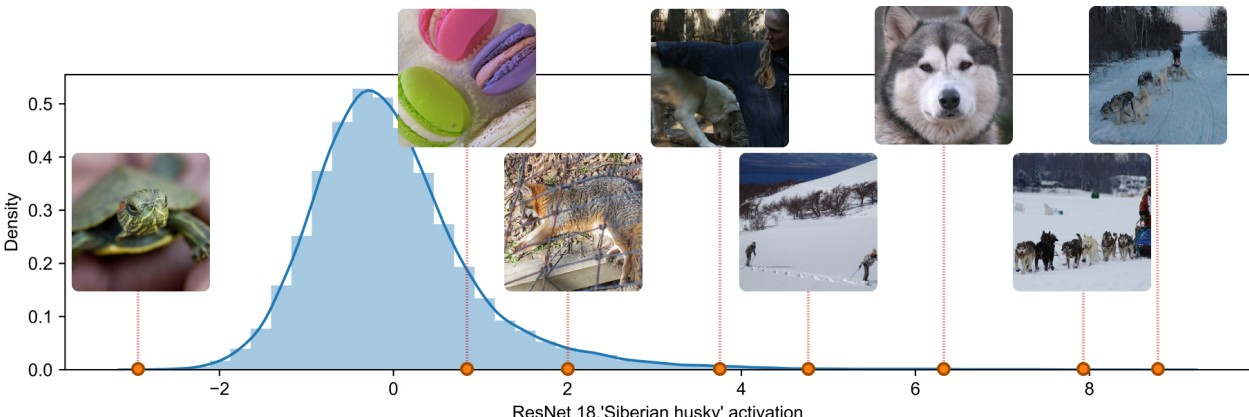

Figure 1: **Distribution of activations for the "Siberian husky" representation.** From the figure we can observe the standardized activation distribution of the "Siberian husky" logit from the ResNet18 model trained on ImageNet. The data was collected across the ILSVRC-2012 validation dataset. Additionally, various input images are dispalyed over their respective activations. Analyzing the activation of representations can provide crucial insights into the behavior of the model. For instance, we observe that the model achieves extremely high activations when there are multiple dogs in the image, corresponding to the "Dogsled" class. However, we also observe a potential spurious correlation, where the model assigns high scores to images with a snowy background.

## 3.1 Data-Aware Extreme-Activation distance

The meaning and semantics of individual representations are frequently characterized and explained by datapoints where these representations exhibit extreme values. This analysis typically concentrates on the most positively activating signals, primarily due to the prevalent use of bounded activation functions like ReLU Glorot et al. (2011). In these functions, positive activation values typically signify the existence of specific patterns within the input signal.

Given the evaluation dataset $D$ and a neural representation $f_i \in \mathcal{F}$, we define a collection of natural Activation-Maximisation signals (n-AMS) as follows:

**Definition 3** (n-AMS). *Let $f_i \in \mathcal{F}$ be a neural representation, and $D = \{x_1, ..., x_N\} \subset \mathbb{D}$ be an evaluation dataset with $N$ datapoints. Assume that the dataset $D$ could be split in $n$ disjoint blocks $D = \bigcup_{i=1}^{n} D_t, D_t = \{x_{td+1}, ..., x_{(t+1)d+1}\}, \forall t \in \{0, ..., n-1\}$ of length $d$.*

*We define a collection of $n$ natural Activation-Maximisation signals (n-AMS) as $S_i = \{s_1^i, ..., s_n^i\}$, where*

$$s_t^i = \arg\max_{x \in D_t} f_i(x), \forall t \in \{0, ..., n-1\}. \tag{6}$$

The suggested definition of n-AMS diverges from the conventional method for explaining the concepts behind the representation by analyzing signals with the highest activations, that is, signals whose activations rank highest across the entire dataset. Instead, n-AMS could be seen as samples from the Extreme Value distribution, thereby allowing for statistical properties to be considered. Additionally, such ditribution is parameterized with parameter $d$, referred to as the *depth*, which represents the size of the subset from which the signal is obtained. Note that we could examine the highest activation signal of the whole dataset by setting $n = 1$ and $d = N$, however, interpreting the representation's semantics by using only one signal might be misleading. Figure 1 illustrates the distribution of activations of the "Siberian husky" logit from the ResNet18 model trained on ImageNet He et al. (2016) across all the images from the ILSVRC-2012 validation dataset Russakovsky et al. (2015), where we can observe that the most activating signal corresponds to the "Dogsled" class. In light of this, we aim to sample several n-AMS from separate data subsets.

We propose that by examining how two neural representations activate each other's n-AMS, we can gain significant insights into the similarity of the learned abstractions. For this, we first introduce the *representation activation vectors* (RAVs).

**Definition 4.** *Let $\mathcal{F} = \{f_1, ..., f_k\}$ be a layer including $k$ neural representations, and $\mathcal{S} = \{S_1, ..., S_k\}$ be a collection of $n$ n-AMS for each of the $k$ representations in the layer. For $\forall i, j \in \{1, ..., k\}$ we define $\mu_j^i = \frac{1}{n} \sum_{t=1}^{n} f_j\left(s_t^i\right)$ as mean activation of $f_j$ given the n-AMS of $f_i$.*

*For any two representations $f_i, f_j \in \mathcal{F}$, we define their pair-wise representation activation vectors (RAVs) $r_{ij}, r_{ji}$ as:*

$$r_{ij} = \begin{pmatrix} \mu_i^i \\ \mu_j^i \end{pmatrix}, \quad r_{ji} = \begin{pmatrix} \mu_i^j \\ \mu_j^j \end{pmatrix}. \tag{7}$$

*In addition, for each neural representation $f_i \in \mathcal{F}$, we define the corresponding layer-wise RAV as follows:*

$$r_{i*} = \begin{pmatrix} \mu_1^i \\ \vdots \\ \mu_k^i \end{pmatrix}. \tag{8}$$

Intuitively, the idea behind RAVs is to capture how one representation's n-AMS are perceived by other representations. RAVs capture the direction of n-AMS signals within two-dimensional vectors when dealing with pair-wise vectors, encoding the information about how two neural representations respond to each other's stimuli. In the layer-wise case, the vectors are $k$-dimensional, utilizing all representations within the layer as descriptors. In practice, to compute RAVs, $n$ n-AMS are gathered for each representation within the layer, inferenced by the model. Then, activations across the representations from the layer are collected and averaged.

To illustrate the concept of Representation Activation Vectors, we calculated n-AMS for five distinct neural representations extracted from the output layer of the ImageNet pre-trained ResNet18 model. These representations corresponded to the classes "Siberian husky", "Alaskan malamute", "Samoyed", "Tiger cat", and "Aircraft carrier", which were selected manually to demonstrate the decreasing visual similarity between the classes and the "Siberian husky" class. Using the ILSVRC-2012 validation dataset, we computed the signals with a sample size of $n = 100$ and a subset size of $d = 500$. Figure 2 presents a scatter plot of activation values across datapoints and pair-wise RAVs. Our results indicate that the angle between these vectors increases with the visual dissimilarity between the classes.

To measure the distance between neural representations that reflect the similarity of learned concepts, we introduce a novel distance metric known as the *Extreme-Activation* distance. This metric assesses the similarity between two neural representations based on the angle between their Representation Activation Vectors. Given that the computation of this distance measure is performed in a *data-aware* mode and relies on the presence of a dataset, we refer to this distance as the natural Extreme-Activation distance or $\text{EA}_n$.

**Definition 5** (Extreme-Activation distance). *Let $f_i, f_j \in \mathcal{F}$ be two neural representations, and $r_{ij}, r_{ji}$ be their pair-wise RAVs. We define a pair-wise Extreme-Activation distance as*

$$d_{EA_n}^p\left(f_i, f_j\right) = \frac{1}{\sqrt{2}} \sqrt{1 - \cos\left(r_{ij}, r_{ji}\right)}, \tag{9}$$

*where $\cos(A, B)$ is the cosine of the angle between vectors $A, B$.*

*Additionally, we define layer-wise Extreme-Activation distance between $f_i, f_j$ as*

$$d_{EA_n}^l\left(f_i, f_j\right) = \frac{1}{\sqrt{2}} \sqrt{1 - \cos\left(r_{i*}, r_{j*}\right)}. \tag{10}$$

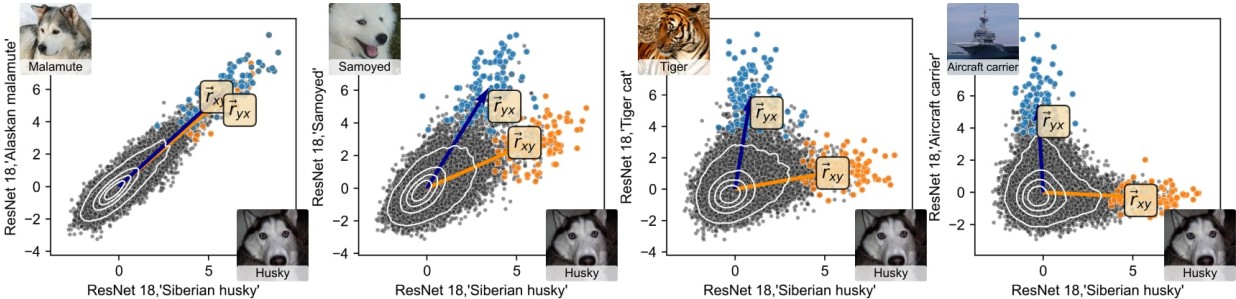

Figure 2: **Joint activation and pair-wise RAVs of different ImageNet representations.** Four scatter plots illustrate the joint activations, n-AMS sets, and pair-wise RAVs for distinct pairs of neural representations. Blue points represent the n-AMS of the representation depicted on the vertical axis, while orange points represent the n-AMS of the representation shown on the horizontal axis. The representations, taken from the ResNet18 output logit layer, include "Alaskan malamute", "Samoyed", "Tiger cat", and "Aircraft carrier", each compared with the "Siberian husky" representation. We can observe that the angle between RAVs reflects the visual similarity between classes, representations were trained to learn: the RAVs of "Siberian husky" and "Alaskan malamute" are almost collinear due to the high visual similarity between the two dog breeds, while the RAVs of "Siberian husky" and "Aircraft carrier" are orthogonal, indicating their visual dissimilarity.

### 3.2 Synthetic Extreme-Activation distance

Although data-aware distance metrics can offer insight into the relationships between representations, their dependence on the data can be viewed as a limitation potentially acting as a bottleneck when analyzing the relationships between a model's internal representations. Modern machine learning models are often trained on closed-source or very large datasets, making it difficult to obtain the exact dataset the model was trained on. If the evaluation dataset, i.e. the dataset utilized for n-AMS sampling lacks concepts that were present in the training data, the resulting n-AMS could potentially be misleading. This is due to the fact that they might not encapsulate the features that the representation has learned to detect, simply because such features are absent in the dataset.

To alleviate the dependence on data, we propose a *data-agnostic* method for computing the Extreme-Activation distance. This approach employs synthetic Activation-Maximization signals, denoted as s-AMS, in place of n-AMS. The s-AMS signals are generated by the model itself through an optimization process, thereby eliminating the need for external generative models or datasets.

**Definition 6** (s-AMS)**.** *Let $f_i \in \mathcal{F}$ be a neural representation. Synthetic Activation-Maximization (s-AMS) signal $\tilde{s}^i$ is defined as a solution to the following optimization problem:*

$$\tilde{s}^i = \arg\max_{\tilde{s} \in \Theta} f_i(\tilde{s}), \tag{11}$$

*where $\Theta$ denotes the set of potential solutions, typically defined by the particular signal parametrization employed for optimization.*

Generating s-AMS for a neural representation is a non-convex optimization problem Nguyen et al. (2019) that typically employs gradient-based methods Erhan et al. (2009); Nguyen et al. (2015); Olah et al. (2017). Starting from a random noise parametrization of input signals, the gradient-ascend procedure searches for the optimal set of signal parameters that maximize the activation of a given representation. Early methods employed standard pixel parametrization Erhan et al. (2009), while modern approaches used Generative Adversarial Network (GAN) generators Nguyen et al. (2016) or Compositional Pattern Producing Networks (CPPNs) Mordvintsev et al. (2018); Stanley (2007). In this study, we use the Feature Visualization method Olah et al. (2017) for s-AMS generation, which parametrizes input signals by frequencies and maps them to the pixel domain using Inverse Fast Fourier Transformation (IFFT). This method is popular for its simplicity

and independence from external generative models, as well as for its ability to be human-interpretable Olah et al. (2020); Goh et al. (2021); Cammarata et al. (2020).

The optimization procedure for s-AMS generation has several adjustable hyperparameters, including the optimization method and transformations applied to signals during the procedure. A key parameter is the number of optimization steps (or epochs) denoted as $m$. This parameter can be seen as analogous to the parameter $d$ used in n-AMS generation. Since different random initializations in the parameter space can lead to the convergence of s-AMS generation into different local solutions, the resulting s-AMS can vary. This variability mirrors that observed when sampling n-AMS. Therefore, analogous to Definition 3, for the representation $f_i$, we introduce a set of s-AMS, comprised of $n$ signals, denoted as $\tilde{S}_i = \{\tilde{s}_1^i, ..., \tilde{s}_n^i\}$.

Figure 3 demonstrates the comparison between s-AMS and most activating signals from the ImageNet dataset for one unit in CLIP ResNet 50 network[†]. The analysis based solely on natural signals leads to erroneous conclusions about the learned concept due to the absence of the true concept in the dataset. As the original training dataset remains undisclosed,

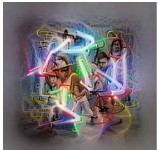
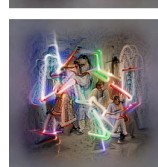
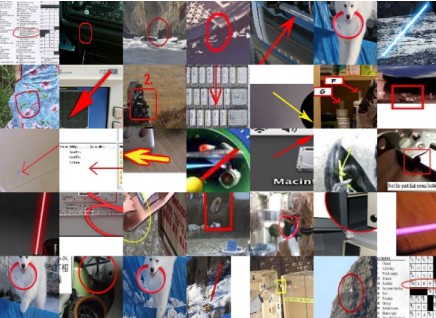

s-AMS          Maximally Activating Images from ImageNet

Figure 3: **Failing to explain "Star Wars" representation with natural images.** Comparison of the s-AMS (left) and most activated images collected from the ImageNet dataset (right) for unit 744 in the last convolutional layer of the CLIP ResNet50 model. Due to the inaccessibility of the training dataset and lack of specific images due to copyright restrictions, natural images do not reveal the underlying concept of the "Star Wars" neuron.

explaining the concepts learned by the representation via identifying the most activating images from ImageNet may lead to misinterpretation, given the absence of "Star Wars"-related images within the ImageNet dataset. In contrast, synthetic Activation-Maximization Signals can depict the learned concepts without any dependency on data.

The *Synthetic Extreme-Activation* distance, or EA$_s$, is defined in a manner analogous to the EA$_n$ distance (Definition 5), with the key distinction being that Representation Activation Vectors are calculated using s-AMS instead of n-AMS. Hence, in contrast to EA$_n$, which evaluates the co-activation of representations based on each other's natural Activation-Maximization signals, EA$_s$ assesses how two representations activate in response to each other's *synthetic* Activation-Maximization signals, i.e., signals generated through an artificial optimization process.

**Definition 7** (Synthetic Extreme-Activation distance). *Let $\mathcal{F} = \{f_1, ..., f_k\}$ be a layer including $k$ neural representations, and $\tilde{\mathcal{S}} = \{\tilde{S}_1, ..., \tilde{S}_k\}$ be a collection of $n$ s-AMS for each of the $k$ representations in the layer. For $\forall i, j \in \{1, ..., k\}$ we introduce synthetic RAVs, by substituting n-AMS with s-AMS in Definition 4: $\tilde{r}_{ij}, \tilde{r}_{ji}$, — pair-wise synthetic RAVs, $\tilde{r}_{i*}, \tilde{r}_{j*}$ — layer-wise synthetic RAVs.*

*We define pair-wise and layer-wise synthetic Extreme-Activation distance between $f_i$ and $f_j$ as*

$$d^p_{EA_s}(f_i, f_j) = \frac{1}{\sqrt{2}}\sqrt{1 - \cos(\tilde{r}_{ij}, \tilde{r}_{ji})}, \quad d^l_{EA_s}(f_i, f_j) = \frac{1}{\sqrt{2}}\sqrt{1 - \cos(\tilde{r}_{i*}, \tilde{r}_{j*})}. \tag{12}$$

Notably, EA$_n$ distance is computed using standardized activations. However, due to the data-agnostic nature of the EA$_s$ distance, standardization cannot be performed without accessing the evaluation dataset; hence, we use the raw representation's activations. Although this could be considered as a limitation, since the EA$_s$ distance is not shift-invariant, we found in our practical experiments that the angles between synthetic and natural RAVs are typically maintained.

---

[†]Signals were obtained from `OpenAI Microscope`.

### 3.3 Properties and Limitations

A key characteristic of the Extreme-Activation distance, applicable to both natural and synthetic contexts, in comparison to other metrics, such as Minkowski, Spearman, and Pearson metrics, is that its computation is based on the activations of two representations given a small subset of AMS, enabling a manual examination of these data points. Such analysis of AMS provides insights into the shared concepts between the two representations. Figure 4 illustrates the procedure of calculating $EA_n$ distance between two representations obtained from the logit layer of the ResNet18 network, specifically for the "Zebra" and "Lionfish" representations. Examination of the angle between Representation Activation Vectors (RAVs) reveals a mutual co-activation between the two representations on each other's n-AMS. By analyzing the signals themselves, which are displayed on the right section of the figure, we can observe the unique shared concepts between these two classes — notably the specific black and white striped pattern exhibited by both animals. Thus, by relying on a limited number of *anchor* data points, the EA distance facilitates the interpretation of the reasons behind the functional similarity between representations.

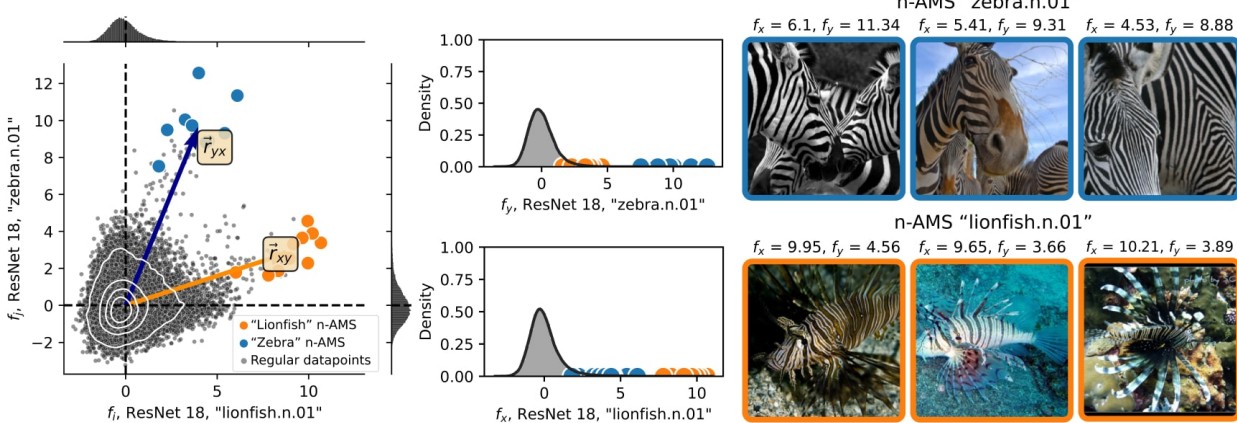

Figure 4: **Explaining the Extreme-Activation Distance.** Figure illustrates $EA_n$ distance between two logit representations from ResNet18 network, corresponding to "Zebra" and "Lionfish" classes. Left figure demonstrates that RAVs experience colinearity, implying mutual co-activation on each other n-AMS ($n = 10, d = 10000$). The center part of the figure presents the activation distributions of representations across the evaluation dataset, where the highlighted points correspond to n-AMS. We can observe that both representations are strongly activated by each other's n-AMS, with the "Lionfish" represented in orange and the "Zebra" in blue. The right-hand portion of the figure presents several n-AMS examples, highlighting the shared feature between the images – the black and white stripe pattern present in both animals – which explains their mutual co-activation.

EA distance, as discussed above, operates on the premise that the representations could be explained by the features in input, that maximally activate them. This viewpoint could be seen as an oversimplification, as it neglects certain attributes that moderately activate or even de-activate representations. Depending on the specific problem and the type of representations in question, such nuances could hold significant importance for understanding the internal decision-making mechanisms of the network. Nonetheless, our experimental findings underline that the EA distance, even when solely focusing on the most activating signals, offers a potent and practical framework for interpreting Deep Neural Networks. This is particularly applicable when examining representations with output confined to the positive realm, for instance, post-ReLU activation function.

One limitation common to all distance metrics between neural representations, including the proposed EA distance, is their inability to account for the *multisemanticity* or *multimodality* Goh et al. (2021) of neural representations  the capacity of a single representation to detect various concepts. In the case of the EA distance, this behavior can be reflected in the high variance of the RAVs, which results from the fact that

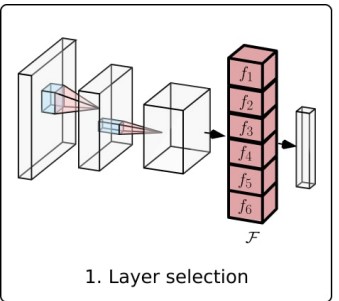 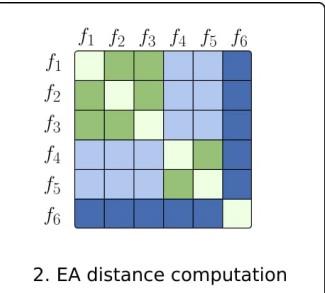 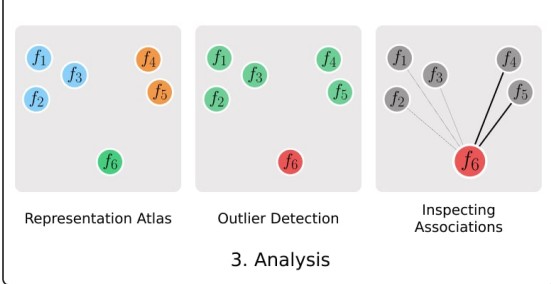

Figure 5: **Outline of the Representation Analysis:** The figure depicts the three fundamental stages of the analysis: 1. Selection of the layer of interest within the specified model. 2. Calculation of EA distances between representations. 3. Analysis of the relationships between representations, encompassing the visualization of representation space, identification of outlier representations, and, when necessary, manual scrutiny of suspicious relationships between representations.

n-AMS originate from multiple modalities, i.e., different concepts. While this remains an open avenue for future research, our work demonstrates that simply averaging activations for RAVs computation provides an effective distance measure.

## 4 Representation Analysis

Traditionally, local XAI methods are used to analyze a model's decision-making process, particularly when assessing potential reliance on unwanted spurious concepts. However, these techniques often fall short when tasked with identifying novel, unfamiliar correlations. In this context, we introduce *Representation Analysis* — a global approach for interpreting a model's decision-making process. This approach is based on the analysis of the internal representations within the models, as well as their interrelationships. After selecting a particular layer, relationships between representations are measured using the proposed EA distance measure. This enables the visualization of the representation space, the analysis of groups of neurons that have learned similar concepts, and the identification of representations that encode anomalous abstractions.

Depending on the problem specifics, the EA distance computation may be executed in a data-aware mode using the $\text{EA}_n$ distance metric or in a data-agnostic mode using the $\text{EA}_s$ distance metric. We denote the latter scenario as **DORA** — *Data-agnOstic Representation Analysis*. The choice between a data-aware and a data-agnostic scenario primarily hinges on the availability of data for analysis. DORA reduces the dependence on data, proving particularly advantageous for interpreting models for which the training data is either unavailable or exceedingly difficult to obtain. Conversely, in practical applications, n-AMS are generally easier to comprehend compared to s-AMS.

Figure 5 provides an overview of the three primary stages of the Representation Analysis pipeline, each of which is further elaborated in the subsequent sections. Initially, a layer of interest is selected from the model. Subsequently, the Extreme-Activation distances between representations are computed. Finally, the relationships between the representations are analyzed. This includes visualizing the representation space via *Representation Atlases*, automatically identifying outlier representations, and, if necessary, conducting a thorough manual examination of the causes behind the relationships between representations. As we will show in the following experiments, these outlier representations often encode undesired concepts.

### Visualizing Representation Spaces with Representation Atlases

Inspired by Carter et al. (2019), the visual examination of the functional diversity within one layer can be done by employing the dimensionality reduction method based on the pre-computed EA distance matrix. Such a visualization, referred to as *representation atlas*, allows researchers to visually examine the topological

landscape of learned representations and identify clusters of semantically similar representations. In the scope of this paper, we employed the widely used UMAP dimensionality reduction algorithm McInnes et al. (2018), which has established itself in recent years as an effective method for visualizing relationships between data points. Figure 6 depicts the representation atlas of the output logit layers of the ResNet18 model trained on ImageNet. Each point in the figure corresponds to an individual neural representation among the 1000 representations in the output layer. The color of each point reflects the WordNet hypernym, a high-level synset, that corresponds to the learned concept of the particular representation. The UMAP visualization, based on the computed EA$_s$ distances, reveals the clusters of semantically similar representations that are preserved, which can be observed in Figure 6.

In comparison with other dimensionality reduction methods, such as t-SNE Van der Maaten and Hinton (2008) and PCA Jolliffe and Cadima (2016), UMAP is scalable, exhibits a faster computation time McInnes et al. (2018); Trozzi et al. (2021); Becht et al. (2019); Wu et al. (2019), and has fewer parameters to tune. Qualitatively, compared to the other methods, UMAP was reported to improve visualizations and accurately represent the data structure on the projected components Trozzi et al. (2021); Becht et al. (2019); Wu et al. (2019).

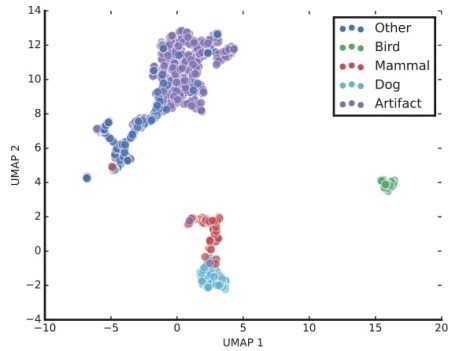

Figure 6: **Representation Atlas of the ResNet18 Logit Layer.** The figure presents a UMAP visualization of the layer-wise EA$_s$ distances between output logit representations from a ResNet18 trained on ImageNet. Each point represents a unique class representation and is color-coded according to its corresponding WordNet hypernym, i.e., the broader category.

## Identifying Outlier Representations

Despite their proven effectiveness across various applications, Deep Neural Networks remain susceptible to learning unintended artifacts and undesired concepts from data. One potential application of our proposed distance measure is the identification of anomalous representations, which deviate from the majority of representations within a layer. This can be achieved by applying Outlier Detection methods Ruff et al. (2021) based on the computed distance matrix. We hypothesize that such anomalies include representations that have learned spurious concepts from the data. In subsequent experiments, we demonstrate the efficacy of the EA distance measure in detecting such outlier representations under controlled conditions. During our practical experiments, we found that while some outlier representations learn unique, task-relevant concepts, these divergent representations often encode undesirable concepts, demonstrating *shortcut learning* or *Clever Hans* behavior.

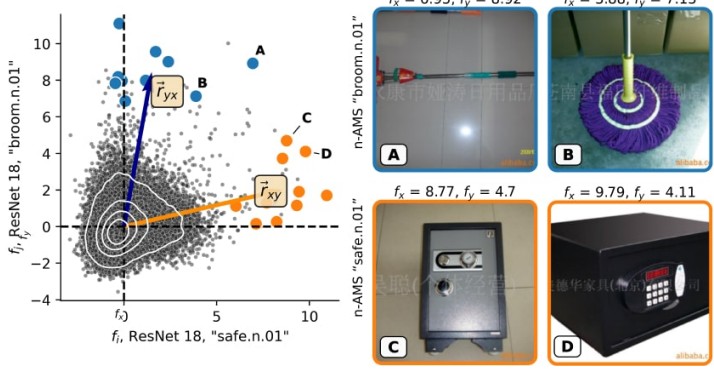

Figure 7: **Functional similarity between representations due to shared watermark dependencies.** Two representations from the ResNet18 model, specifically "Broom" and "Safe", show significant co-activation on some n-AMS, despite their visual dissimilarity, which would typically suggest orthogonal RAVs. A closer inspection of the mutual n-AMS reveals that the common feature is the presence of watermarks on the images, which is shown among 4 n-AMS, mutually activating both representations.

## Investigating Individual Relationships

In order to determine the cause behind a specific EA distance, individual relationships between representations can be examined manually. This can be accomplished through visualizing RAVs and AMS, as discussed

in Section 3.3. For instance, Figure 7 illustrates the spurious correlation between the "Broom" and "Safe" representations from the output logit layer of ResNet18, which accounts for the low EA distance between these representations. Upon examining the shared n-AMS, it becomes clear that the n-AMS, which maximally activate both representations, contain a single common feature watermarks. Such visualizations can aid in investigating the underlying functional similarity between representations and offer insight into the reasons behind such similarities.

## 5 Evaluation

The practical utility of a distance metric between representations fundamentally depends on its capacity to gauge the similarity between concepts that various model representations have learned. The manner in which this similarity is assessed can take on multiple forms, but from a standpoint of explainability, it is important that the distance metric is aligned Muttenthaler et al. (2022); Gabriel (2020) with human perception - the computed distances should resonate with our human senses of similarity and difference. Assuming that we know what abstractions two representations are detecting, an effective distance metric should label representations that detect concepts perceived as similar by humans as alike, and those identifying concepts perceived as distinct by humans as dissimilar. Such attributes of the distance metric facilitate the clustering of representations that are conceptually similar and the identification of outlier representations that encode concepts anomalous to the task at hand. This has advantages as it aids in revealing shortcuts and Clever Hans effects, which often appear as out-of-distribution anomalies and are unnatural to the assigned task - for example, textual watermarks in the context of object classification Lapuschkin et al. (2019) or specific tokens in medical image classification Geirhos et al. (2020).

To quantitatively evaluate the alignment, we compared human-defined semantic distances between concepts, which we refer to as *semantic baselines*, with distance matrices computed between representations trained to learn these concepts. For our study, we utilized two prevalent computer vision datasets, namely ILSVRC-2012 Russakovsky et al. (2015) and CIFAR-100 Krizhevsky (2009). We established a measure of distance between concepts by utilizing semantic distances between the class labels. This was accomplished by mapping the classification labels to entities within the WordNet taxonomy database Miller (1995), a lexical database that organizes English words into a taxonomy of synonym sets, or synsets. In this taxonomy, each synset represents a group of words that are synonyms or have the same meaning. WordNet organizes these synsets into a hierarchy, with more specific concepts being nested under more general ones.

For the ImageNet dataset, class labels were automatically mapped to the corresponding WordNet synsets due to their inherent linkage, whereas for the CIFAR-100 dataset, the labels were manually matched to their respective synsets. It is important to note that semantic distance does not directly equate to the visual similarity between concepts. However, positive correlations between semantic and visual similarities have been reported, thereby demonstrating a significant positive relationship between semantic and visual distances Deselaers and Ferrari (2011); Brust and Denzler (2019).

Given the WordNet taxonomy in a form of a graph $\mathcal{G} = (V, E)$ with root $r \in V$, the baseline semantic distances between entities from the WordNet database were computed using the following three distance measures:

- **Shortest-Path distance**

  Given two vertices $c_i, c_j \in V$ the distance between vertices is determined by the length of the shortest path that connects the two entities in the taxonomy

  $$d_{SP}(c_i, c_j) = l(c_i, c_j).$$

  where $l(c_i, c_j)$ is the function that returns the minimal number of edges that need to be traversed to get from $c_i$ to $c_j$.

- **Leacock-Chodorow distance Leacock and Chodorow (1998)**

Given two vertices $c_i, c_j \in V$ the distance between vertices is determined by a logarithm of the shortest-path distance with additional scaling by the taxonomy depth:

$$d_{LC}(c_i, c_j) = \log \frac{l(c_i, c_j) + 1}{2T} - \log \frac{1}{2T},$$

where $T = \max_{c \in V} l(r, c)$ is the taxonomy depth.

- **Wu-Palmer distance Wu and Palmer (1994)**

  Given two vertices $c_i, c_j \in V$ the Wu-Palmer distance is defined as:

$$d_{SP}(c_i, c_j) = 1 - 2\frac{l(r, lcs(c_i, c_j))}{l(r, c_i) + l(r, c_j)},$$

  where $lcs(c_i, c_j)$ is the Least Common Subsumer Pedersen et al. (2004) of two concepts $c_i$ and $c_j$.

Furthermore, we have utilized the textual labels from both ImageNet and CIFAR100 datasets and calculated the Word2Vec Mikolov et al. (2013) similarity between class labels.

- **Word2Vec distance**

  Given textual labels $t_i, t_j$ of two concepts $c_i, c_j$, we define Word2Vec distance as

$$d_{W2V} = \frac{1}{\sqrt{2}}\sqrt{1 - \cos_{W2V}(t_i, t_j)},$$

  where $\cos_{W2V}(A, B)$ is the cosine of the angle between Word2Vec embeddings of the words $A, B$.

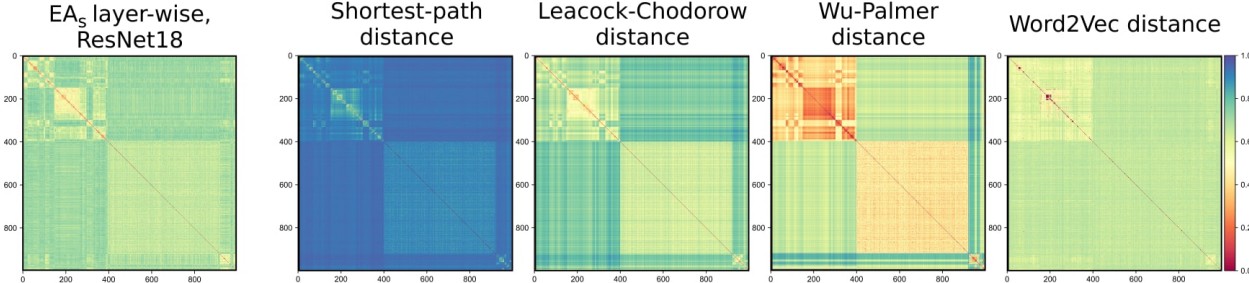

Figure 8: **Comparison between EA$_s$ distance matrix and Seamntic Baselines**: From left to right: the EA$_s$ distance metric computed for the output logits of the ImageNet pre-trained ResNet18 model, Shortest-Path, Leacock-Chodorow, Wu-Palmer distances from WordNet taxonomy, and Word2Vec distance.

Figure 8 illustrates the comparison between the functional EA$_s$ distance matrix, derived from 1000 representations from an ImageNet-trained ResNet18 network, and semantic baselines  these are the human-defined distances between ImageNet concepts. To evaluate the alignment between the proposed distance metric and human-defined baselines, we employed the Mantel Test Mantel (1967), which is often applied in ecology and evolutionary biology to measure the correlation between two distance matrices. The test calculates the correlation coefficient $\rho$, which indicates the strength of the relationship between the two matrices, and the $p$-value of the test, which describes the statistical significance of the correlation.

It is essential to note that while we evaluate the alignment based on human-defined semantic benchmarks, optimizing such metrics should not be the ultimate objective when proposing new distance metrics between representations. This is because DNNs can naturally employ different decision-making strategies than humans, and these differences may not always be attributed to spurious correlations. For instance, taxonomy-based approaches might be sub-optimal compared with attributing freedom to the models to train for the desired tasks Binder et al. (2012). Conversely, in Computer Vision, network representations are expected to be aligned to some extent due to the correlations between the visual and semantic similarity of classes.

### 5.1 Hyperparameter selection

This section examines the selection of parameters in terms of their ability to attain optimal alignment with the semantic baselines. In our empirical analysis, we utilized a pre-trained ResNet18 model on ImageNet, along with the ILSVRC-2012 validation set consisting of 50,000 images and 1,000 classes, employed for the data-aware metrics. Herein, we calculated the distance metrics between output logit representations, that is, the pre-softmax representations. For data-aware metrics, outputs of representations underwent normalization as discussed in Section 3. Conversely, for the data-agnostic distance metric, specifically $EA_s$, no normalization procedure was undertaken.

#### Minkowski distance

To investigate how different values of the parameter $p$ affect the coherence to semantic baselines, we varied the parameter and evaluated the alignment with four semantic baselines for each case. Figure 9 shows the effect of parameter selection on the Mantel test statistic. We observed that the optimal average value of the statistic across the four baselines was achieved for $p = 2$. However, for future experiments, we selected the second-best parameter choice with $p = 1$ due to the natural connection between Euclidean distance and Pearson correlation. We also observed that higher values of $p$ generally result in lower alignment, possibly due to sensitivity to the large amplitudes of individual datapoints.

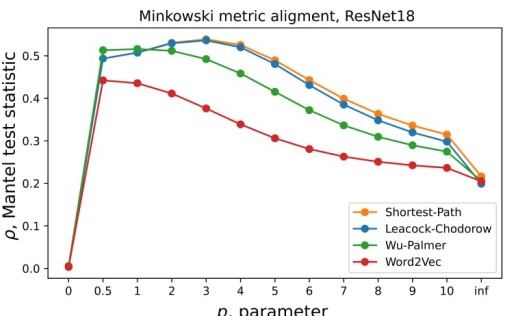

Figure 9: **Impact of Parameter Selection in Minkowski Distance on Alignment with Semantic Baselines.** To assess the alignment with respect to the four semantic baselines, we calculated the Minkowski distance on the output logits of the ResNet18 network while varying the parameter $p$. The Mantel correlation statistic is reported for each semantic baseline at each parameter value.

#### $EA_n$ distance

$EA_n$ is influenced by two key parameters: $n$, which denotes the number of n-AMS signals gathered, and $d$, which represents the size of the subset collected from each signal. To investigate the impact of parameter selection, we varied these parameters for both pair-wise and layer-wise modes. Figure 10 shows the average Mantel correlation statistic across four semantic baselines for each hyperparameter choice. Our observations reveal that, in general, increasing the number of collected n-AMS, irrespective of the parameter $d$, has a positive impact on the alignment. However, the optimal depth $d$ is achieved when n-AMS are taken from subsets of $d = 50$ datapoints.

#### $EA_s$ distance

In the data-agnostic version of the Extreme-Activation distance, the choice of hyperparameters depends on the s-AMS generation method used. In our study, we employed the Feature Visualisation method to generate s-AMS, and we identified two critical hyperparameters: $n$, which is the number of generated s-AMS per representation, and $m$, which is the number of optimization epochs per signal. Figure 11 depicts the impact of the $EA_s$ distance measure's hyperparameter selection on semantic baseline alignment. We observed that while increasing the number of generated s-AMS generally has a positive effect, this effect is negligible compared to the positive impact of increasing the number of optimization epochs per representation. This is likely due to the generation algorithms' convergence to better local optima, resulting in improved visual preciseness of the images, as illustrated on the right side in Figure 11.

### 5.2 $EA_s$ Preserves the Angles Between Natural Representation Activation Vectors

Although both n-AMS and s-AMS activate specific neural representations maximally, the adversarial nature of synthetic signals needs to be considered. In our experiments, we observed that while the generated s-AMS

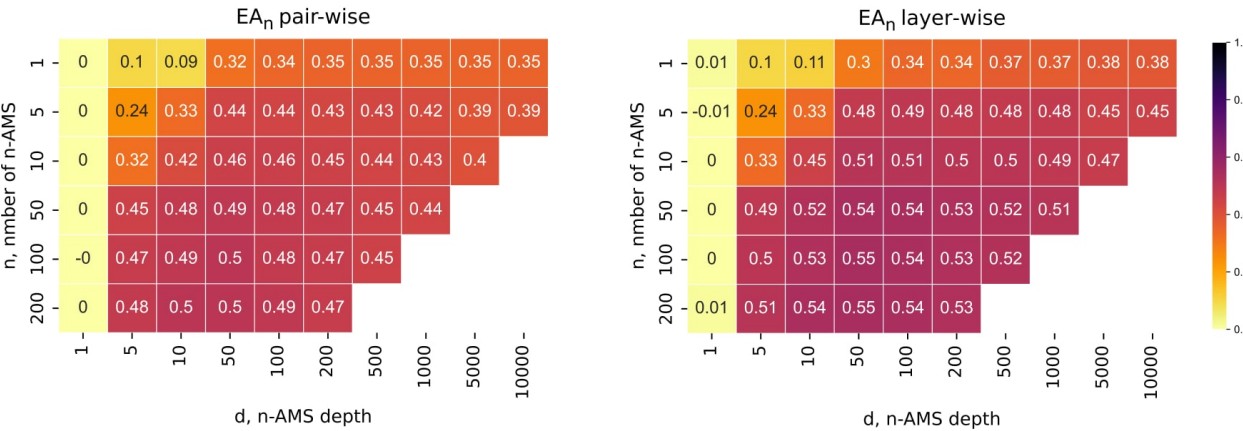

Figure 10: **Impact of parameter selection in $EA_n$ distance on alignment with semantic baselines.** To assess alignment with four semantic baselines, we calculated data-aware EA distance on the output logits of the ResNet18 network while varying the parameters $n$ and $d$ for both pair-wise and layer-wise options. The average Mantel correlation statistic across four semantic baselines is reported at each cell.

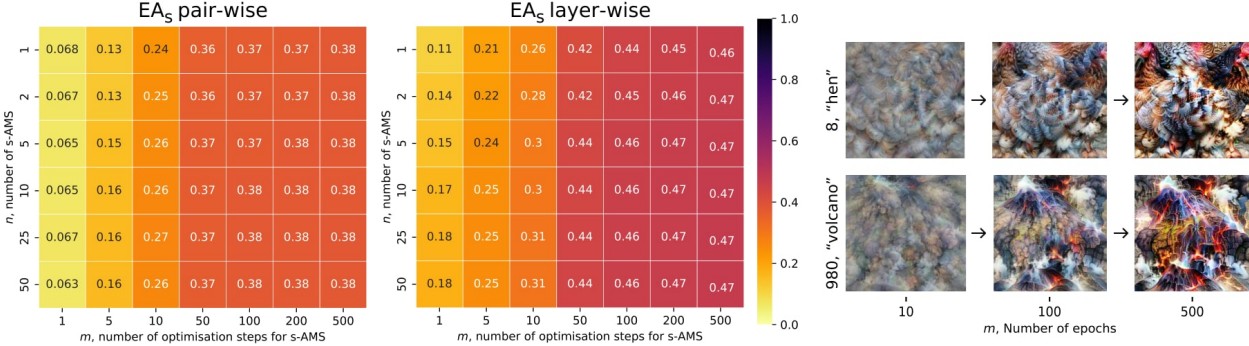

Figure 11: **Impact of the parameter selection in $EA_s$ distance on alignment with semantic baselines.** To evaluate the alignment between the $EA_s$ distance and four semantic baselines, we computed the data-agnostic $EA_s$ distance using the ResNet18 network's output logits while varying the hyperparameters $n$ and $m$ for both pair-wise and layer-wise options. For each cell, we reported the average Mantel correlation statistic across the four semantic baselines. The effect of the hyperparameter $m$, which corresponds to the number of optimization steps taken for s-AMS generation, on two neural representations from the ResNet18 output logit layer is shown on the right.

are far from the original *natural* image domain, the angles between natural and synthetic RAVs are consistent, providing additional evidence to the utility of the $EA_s$ distance metric.

To evaluate the angle conservation quantitatively, we employed a ResNet18 pre-trained on the ImageNet dataset and computed $EA_n$ and $EA_s$ distances between the output logit representations, i.e., all 1000 ImageNet classes. It's important to mention that the $EA_n$ distance is calculated over normalized representations, while $EA_s$ is based on the unnormalized output of representations. In this experiment, we kept the number of signals constant at $n = 50$ for both distance metrics, yet varied the parameter $d$ for n-AMS generation and the parameter $m$ for s-AMS generation. Given $\mathcal{F} = \{f_1, \ldots, f_k\}$, which corresponds to the ResNet18 output layer with $k = 1000$ neural representations, we evaluated the Root Mean Square Error (RMSE) between pairwise $EA_n$ and $EA_s$ distances:

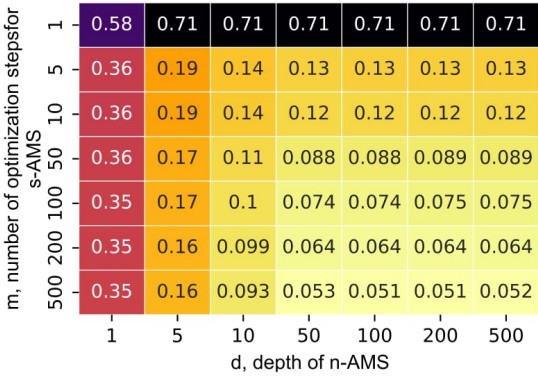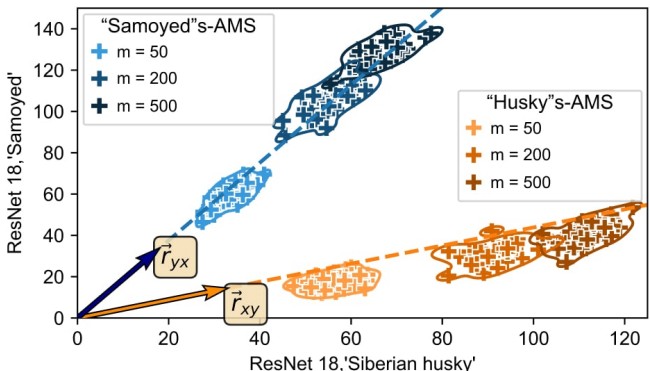

Figure 12: **Similarity and angle preservation between EA$_n$ and EA$_s$ distance measures.** The left part of the figure shows the RMSE (lower is better) between pair-wise EA$_n$ and EA$_s$ distances on the output layer of the ResNet18 network, with a fixed parameter $n = 50$ for both metrics, while varying parameters $d$, corresponding to the subset size in n-AMS sampling, and $m$, number of epochs for s-AMS generation. On the right part of the figure, the distributions of pair-wise activations of s-AMS signals are visualized with different parameters $m$ for two neural representations, namely "Samoyed" and "Siberian husky", overlayed with the direction of natural RAVs computed with $n = 50$ and $m = 1000$. The length of natural RAVs was extended to enhance visibility.

$$RMSE = \sqrt{\frac{\sum_{i=1}^{k}\sum_{j=i+1}^{k}\left(d_{EA_n}^{p}(f_i, f_j) - d_{EA_s}^{p}(f_i, f_j)\right)^2}{k(k-1)/2}}, \tag{13}$$

where $k(k-1)/2$ corresponds to the number of all unique pairs of two different functions from a set of $k$ functions.

Figure 12 illustrates the similarity between the computed EA$_n$ and EA$_s$ distances between the representations of the 1000 ImageNet classes. In the left part of the figure, which shows RMSE between the two distance measures for different parameters, we observe that for each parameter $m$ for EA$_s$ distance, the lowest error is achieved with an EA$_n$ distance with high values of $d$. This suggests that the EA$_s$ distance accurately captures the angle between RAVs corresponding to the images with the highest activation percentile. Additionally, we observed that increasing the parameter $m$ is beneficial to lowering the RMSE between natural and synthetic measures. Furthermore, the right part of the figure shows the direction of natural RAVs and activations of s-AMS for "Samoyed" and "Siberian husky" representations from ResNet18. From this figure, we can observe that the angle between natural RAVs and synthetic RAVs is conserved.

### 5.3 Evaluating the alignment with human judgment

In this experiment, we quantitatively assess the alignment of the discussed distance metrics with the human-defined distance measures across different datasets and architectures. To this end, we employed eight different architectures for two datasets, ImageNet and CIFAR100. For ImageNet, we employed ResNet18 He et al. (2016), AlexNet Krizhevsky et al. (2017), ViT Dosovitskiy et al. (2020), BEiT Bao et al. (2021), Inception V3 Szegedy et al. (2016), DenseNet 161 Huang et al. (2017), MobileNet V2 Sandler et al. (2018), ShuffleNet V2 Ma et al. (2018), while for CIFAR-100, we used ResNet 18, ResNet 9, MobileNet V2, ShuffleNet V1, and V2, as well as NASNet Qin and Wang (2019), SqueeeNet Iandola et al. (2016) and VGG 11 Simonyan and Zisserman (2014).

We computed functional distances with optimal hyperparameters found in Section 5.1, including Minkowski $p = 1$, Pearson, Spearman, EA$_n$ with $n = 50, d = 200$, and EA$_s$ with $n = 3, m = 500$, on the output logit layer for each model. We then compared each distance matrix with four semantic baselines: Shortest-Path, Leacock-Chodorow, Wu-Palmer distances from WordNet taxonomy, and Word2Vec distance. This comparison

Table 1: **Alignment of Distance Metrics in ImageNet Trained Models**: Each cell represents the average Mantel test statistic (higher is better) across four semantic baselines: Shortest-Path, Leacock-Chodorow, Wu-Palmer distances, and Word2Vec distance. All results demonstrate statistical significance with $p < 0.001$.

| | Minkowski $p = 1$ | Pearson | Spearman | $EA_n$ | | $EA_s$ | |
| --- | --- | --- | --- | --- | --- | --- | --- |
| | | | | p-w | l-w | p-w | l-w |
| ResNet18 | 0.49 | 0.50 | 0.48 | 0.49 | 0.55 | 0.38 | 0.47 |
| BeIT | 0.32 | 0.36 | 0.29 | 0.44 | 0.50 | 0.39 | 0.47 |
| MobilenetV2 | 0.46 | 0.46 | 0.45 | 0.47 | 0.52 | 0.40 | 0.50 |
| DenseNet161 | 0.46 | 0.47 | 0.44 | 0.49 | 0.54 | 0.32 | 0.39 |
| ShuffleNetV2 | 0.21 | 0.21 | 0.19 | 0.29 | 0.30 | 0.19 | 0.16 |
| InceptionV3 | 0.31 | 0.34 | 0.32 | 0.38 | 0.49 | 0.22 | 0.27 |
| AlexNet | 0.52 | 0.53 | 0.52 | 0.52 | 0.55 | 0.42 | 0.45 |
| ViT | 0.53 | 0.54 | 0.52 | 0.54 | 0.58 | 0.48 | 0.53 |
| **Mean** | **0.41** | **0.43** | **0.40** | **0.45** | **0.50** | **0.35** | **0.40** |

Table 2: **Alignment of Distance Metrics in CIFAR100 Trained Models**: Each cell represents the average Mantel test statistic (higher is better) across four semantic baselines: Shortest-Path, Leacock-Chodorow, Wu-Palmer distances, and Word2Vec distance. All results demonstrate statistical significance with $p < 0.001$.

| | Minkowski $p = 1$ | Pearson | Spearman | $EA_n$ | | $EA_s$ | |
| --- | --- | --- | --- | --- | --- | --- | --- |
| | | | | p-w | l-w | p-w | l-w |
| ResNet9 | 0.32 | 0.37 | 0.33 | 0.41 | 0.52 | 0.27 | 0.30 |
| ShuffleNetV2 | 0.49 | 0.52 | 0.49 | 0.53 | 0.59 | 0.43 | 0.47 |
| MobileNetV2 | 0.50 | 0.51 | 0.49 | 0.52 | 0.59 | 0.40 | 0.44 |
| ResNet18 | 0.43 | 0.47 | 0.45 | 0.48 | 0.57 | 0.30 | 0.37 |
| ShuffleNet | 0.48 | 0.51 | 0.49 | 0.52 | 0.58 | 0.42 | 0.46 |
| VGG11 | 0.30 | 0.31 | 0.31 | 0.36 | 0.43 | 0.23 | 0.23 |
| NasNet | 0.48 | 0.51 | 0.48 | 0.52 | 0.59 | 0.36 | 0.41 |
| SqueezeNet | 0.50 | 0.52 | 0.51 | 0.53 | 0.59 | 0.45 | 0.51 |
| **Mean** | **0.44** | **0.46** | **0.44** | **0.48** | **0.56** | **0.36** | **0.40** |

yielded four Mantel test statistics per distance metric. The results of the evaluation are presented in Table 1 for ImageNet-trained models and in Table 2 for CIFAR100 models, where we averaged the four Mantel correlation test statistics for each model and distance metric. Our analysis indicates that the layer-wise $EA_n$ metric's distance is generally more favorable due to its stronger linear relationship with all four baseline metrics. Furthermore, we observed that the data-agnostic $EA_s$ metric is on par with data-aware metrics in terms of alignment with the semantic baselines.

## 5.4 Evaluating Anomaly-Identification capabilities

The alignment of distance metrics between neural representations and human judgment of concepts presents an intriguing potential application. Specifically, we can identify representations that are semantically anomalous compared to the majority of learned representations, based on the functional distance. While these representations may simply learn unique individual concepts, we demonstrate in further experiments that in real-life scenarios they might correspond to the undesired concepts from spurious correlations in the training data that diverge from the typical (intended) decision-making strategy.

To assess the usefulness of the alignment between distance metrics and human-defined semantic baseline, we conducted the experiment, where we measured the ability of the distance metrics to detect anomalous

representations. For this purpose, we trained a ResNet18 network on a combination of two conceptually different datasets. The combined dataset comprised the Tiny Imagenet Le and Yang (2015), containing 200 ImageNet classes, and the MNIST handwritten-numbers dataset Deng (2012), containing 10 handwritten numbers, resulting in a total of 210 classes. MNIST images were upsampled to the size of $3 \times 64 \times 64$ pixels to match the size of images in Tiny ImageNet. After training on the combined dataset in the image classification task, we computed functional distances between the output logits and evaluated the ability of different Outlier Detection (OD) methods to detect MNIST logits, given the computed distance matrices only. For this, we utilized five different Outlier Detection methods: the Angle-based Outlier Detector (ABOD) Kriegel et al. (2008), Feature Bagging (FB) Lazarevic and Kumar (2005), Isolation Forest (IF) Liu et al. (2008), Local Outlier Factor (LOF) Breunig et al. (2000) and One-class SVM (OCSVM) Schölkopf et al. (2001). We evaluated the performance of the Outlier Detection methods using the AUC ROC metric for the binary classification between Tiny ImageNet and MNIST representations. To ensure stability in light of the stochastic nature of some outlier detection methods, the results of the outlier detection were repeated 100 times with different random states.

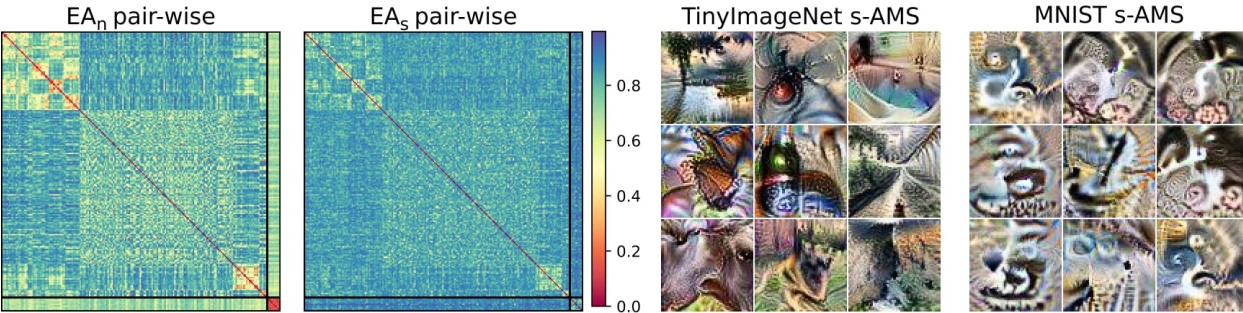

Figure 13: **Anomaly-Detection Evaluation Experiment.** From left to right, pair-wise $\text{EA}_n$ distance matrix between the output logits of the network trained on the combined dataset, the $\text{EA}_s$ distance matrix, s-AMS for Tiny ImageNet logits, and s-AMS for MNIST logits. MNIST representations are highlighted on both distance matrices in the bottom right corner, revealing a block structure in both distance metrics that suggests a high degree of functional differences between Tiny ImageNet representations and semantically distinct MNIST representations. On the left, we can visually observe differences between the s-AMS of Tiny ImageNet and MNIST representations.

We utilized the same hyperparameter configuration for distance computation as described in Section 5.3. The effectiveness of EA distances, both natural and synthetic, in distinguishing between representations of Tiny ImageNet and MNIST is demonstrated in Figure 13, as evidenced by the block structure of the distance matrices. This behavior can be attributed to the visual dissimilarities between the classes, where Tiny ImageNet classes exhibit natural and diverse features that are typical for natural images, while MNIST images consist of white digits on a black background. In the case of synthetic EA distance, the ability to detect MNIST representations is based on the visual differences in the s-AMS, which are depicted in the right-hand portion of Figure 13. The s-AMS-based EA distance measure depends on the network's ability to perceive self-generated s-AMS, and we can observe distinct dissimilarities between the patterns of s-AMS for Tiny ImageNet classes, which contain high-level natural concepts, and the more data-specific patterns for MNIST classes, which illustrate the network's perception of white-on-black handwritten digits and letters.

The results of the described experiment are presented in Table 3, which indicate that, in general, all distance metrics are capable of detecting MNIST representations. However, the EA distance metrics are generally more effective in detecting semantically artifactual representations, whereas the pairwise $\text{EA}_n$ metric is the most effective.

Table 3: **Anomaly Identification Performance of Distance Metrics.** The table displays the average AUC ROC (higher is better) binary classification performance of the Outlier Detection methods across 100 re-trials, in the task of detecting MNIST representations among the combined Tiny ImageNet and MNIST representations, specifically in the output layer of the trained network.

|  | *Minkowski* | *Pearson* | *Spearman* | *$EA_n$* | | *$EA_s$* | |
|---|---|---|---|---|---|---|---|
|  | $p = 1$ | | | *p-w* | *l-w* | *p-w* | *l-w* |
| *ABOD* | 0.56 | 0.63 | 0.58 | 0.91 | 1.00 | 0.82 | 0.71 |
| *FB* | 0.97 | 0.99 | 0.81 | 1.00 | 1.00 | 0.89 | 0.87 |
| *IF* | 0.83 | 0.87 | 0.64 | 0.94 | 0.70 | 0.76 | 0.61 |
| *LOF* | 0.65 | 0.53 | 0.55 | 0.67 | 0.96 | 1.00 | 0.87 |
| *OCSVM* | 1.00 | 1.00 | 0.95 | 1.00 | 0.67 | 1.00 | 0.72 |
| **Mean** | **0.80** | **0.80** | **0.71** | **0.90** | **0.87** | **0.89** | **0.76** |

# 6 Experiments: Finding Outlier Representations with DORA

In this section, we illustrate the broad applicability of the DORA framework and demonstrate that outlier representations, often found in intermediate layers, can frequently encode malicious and undesirable concepts.

## 6.1 ImageNet pre-trained networks

Pre-trained networks on ImageNet have become an essential component in the field of Computer Vision. Their capability to recognize a diverse set of objects and scenes makes them particularly useful as a starting point for a wide range of computer vision tasks. They are frequently utilized for fine-tuning to specific tasks or as a feature extractor, where the images are encoded by the networks for further computations Zhuang et al. (2020); Weiss et al. (2016).

In the following, we explore the feature extractor representations of three widely-used pre-trained models: ResNet18 He et al. (2016), MobileNetV2 Sandler et al. (2018), and DenseNet121 Huang et al. (2017). Using LOF outlier detection, we found latent layers with representations that appear to be watermark detectors, e.g., detecting Chinese and Latin text patterns. As ImageNet does not have a specific category for watermarks, these representations could be seen as Clever-Hans artifacts and deviate from desired decision-making Lapuschkin et al. (2019); Anders et al. (2022). To verify these representations can detect watermarks, we created two binary classification datasets, for Chinese and Latin watermarks, containing normal images and identical images, with inserted random watermarks, evaluating the sensitivity of individual representations using the AUC ROC classification measure. To ensure the detection of characters and not specific words/phrases (unlike CLIP models Goh et al. (2021)), the probing datasets were generated with random characters (for more details we refer to the Appendix). Our results show that not only the reported outliers but also neighboring representations in EA distance are affected by artifactual behavior. Lastly, we find that this behavior persists during transfer learning, posing a risk for safety-critical fields like medicine.

**ImageNet ResNet18**

We applied DORA to analyze the Average Pooling layer of the ResNet18 model, which consists of the $k = 512$ high-level representations that are commonly used without further modification during transfer learning. Following the DORA approach, we calculated $EA_s$ layer-wise distance with $n = 5$ s-AMS per each representation and with $m = 500$, based on our findings in the section 5.1. After calculating the $EA_s$ distances, we used the LOF method with a contamination parameter $p = 0.01$ (corresponding to the top 1% of representations), and the number of neighbors was set to 20.

LOF identified five outlier representations, namely neurons 7, 99, 154, 160, 162, and 393. The outlier neuron 154, displayed a specific, recognizable pattern in s-AMS that could be perceived as the presence of Chinese logograms. By probing the network on a binary classification problem between images watermarked with Chinese logograms vs normal images, Neuron 154 showed the highest detection rate (AUC ROC of

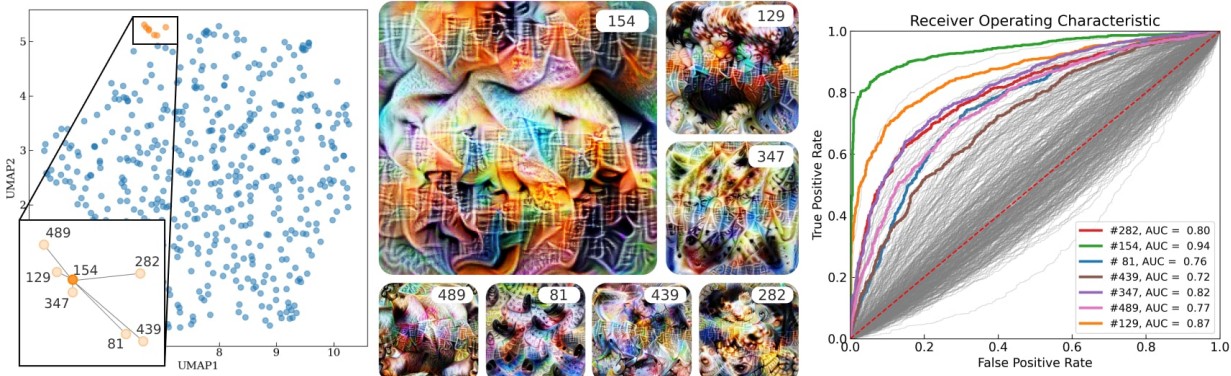

Figure 14: **Cluster of Clever-Hans representations in the ResNet18 feature extractor.** From left to right: representation atlas of the ResNet18 average pooling layer with the highlighted cluster of Clever-Hans representations (left), s-AMS of the representations in the cluster (middle), and AUC ROC sensitivity scores for the detection of images with Chinese watermarks in the binary classification problem (right), where colored curves correspond to the behavior of representations in the cluster and gray curves for other representations. From the s-AMS of neuron 154, we can observe symbolic patterns resembling Chinese logograms learned by the neuron as well as by its closest neighbor neurons. We can observe that the outlier neuron 154 exhibits the highest AUC value (green curve), followed by its nearest neighbors.

0.94) towards the class with watermarked images, providing significant evidence that this representation is susceptible to the Clever-Hans effect. Further analysis of neighboring representations in $\text{EA}_s$ distance showed that they also exhibit similar behavior. The results of the analysis of the ResNet 18 average pooling layer are shown in Figure 14, illustrating the cluster of Clever-Hans representations found, along with their s-AMS and AUC ROC performance on the binary classification problem. Additional information on the dataset generation and the identified outlier representations can be found in the Appendix. Note that in general, the presence of such artifacts could indeed pose serious risks and may lead to a degradation in classifier performance (see Anders et al. (2022)).

In the further investigation of the model, we inferenced s-AMS signals of representations in the reported CH-cluster and obtained their predictions by the model. Among the selected signals, the model predominantly predicted an affiliation of these signals with the classes "carton", "swab", "apron", "monitor" and "broom", which is in line with the reported spurious correlation of the "carton" class and Chinese watermarks Li et al. (2022). Upon computing the corresponding s-AMS signals for these logits, we were able to confirm their association with CH-behaviour, as they displayed clear, visible logographic patterns, specific to Chinese character detectors, in their corresponding s-AMS. Corresponding signals and additional information could be found in Appendix.

**ImageNet MobileNetV2**

We used DORA with the same parameters as in the previous experiment ($n = 5$ s-AMS per each representation and $m = 500$ epochs for s-AMS generation) to analyze the "features" layer of MobileNetV2 network Sandler et al. (2018), which consists of $k = 1280$ channels with $7 \times 7$ activation maps. The analysis was performed on channels by averaging the resulting activation maps of neurons. We calculated the EA distances between representations and applied the LOF method with a contamination parameter of 0.01 which yielded 13 outlier representations. Upon visual inspection of the s-AMS of these representations, we observed distinct patterns specific to Chinese character detectors in neurons 397, 484, 806, and 1131. Figure 15 illustrates the s-AMS of these neurons, as well as the sensitivity of neurons in the Chinese-character detection task. We can observe that the neighbors of these neurons (397, 484, 806, 1131) are sensitive to CH artifacts and form a distinctive cluster visible in the representation atlas.

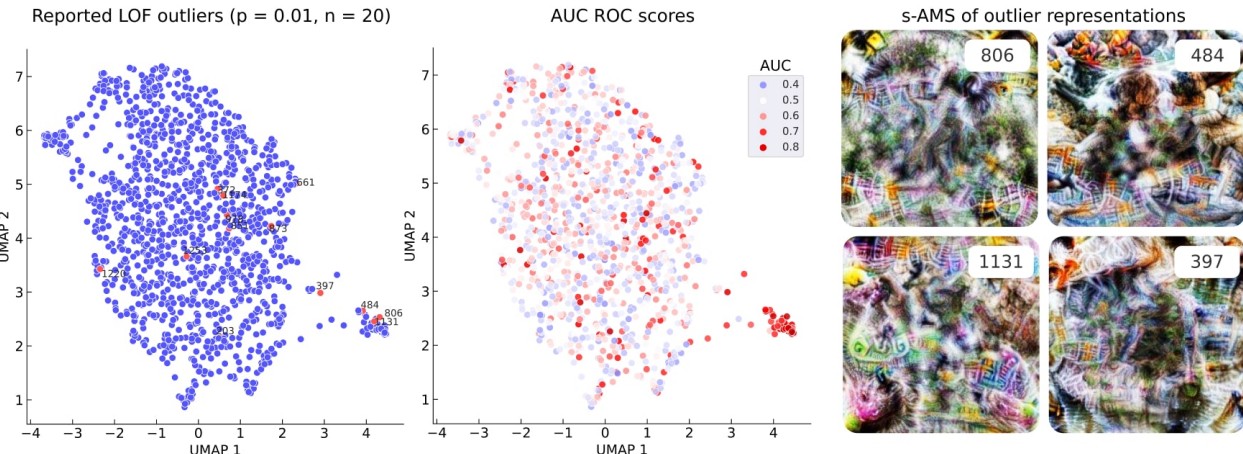

Figure 15: **Cluster of Clever-Hans representations in the MobileNet V2 feature extractor.** The left figure illustrates the outlier representations as identified by the LOF OD method, overlaid on the DORA representation atlas. The middle figure displays the sensitivity of the neural representations to Chinese watermarks, where the highly-sensitive cluster of neurons can be clearly observed in the bottom-right part of the atlas, including 3 reported outlier representations. The right graph illustrates the s-AMS of several of the reported outlier neurons, which exhibit a distinctive logographic pattern typical of Chinese character detectors.

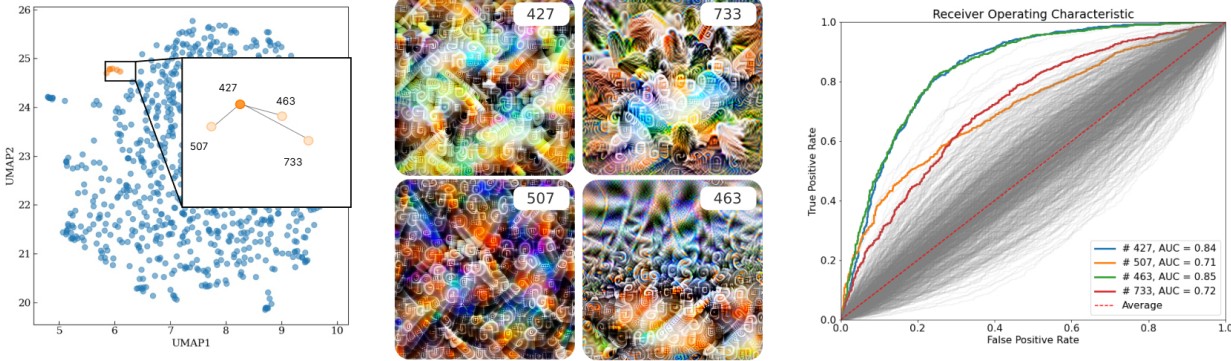

Figure 16: **DenseNet121 — Latin text detector.** Applying DORA to the last layer of the feature extractor of DenseNet121 yields, among others, Neuron 427 as an outlier, which corresponds to the upper left of the 4 feature visualizations. From neuron 427 as well as from its three closest neighbors (shown left), we can observe semantic concepts resembling Latin text characters. The AUC values were computed using the average channel activations on the Latin probing dataset. As shown, the AUCs are high for the representation outliers found by DORA, compared to most of the other representations, which indicates that they indeed learned to detect Latin text patterns.

## ImageNet DenseNet 121

We conducted a similar analysis on the last layer of the feature extractor of the ImageNet pre-trained DenseNet121 model, which consists of $k = 1024$ channel representations with $7 \times 7$ activation maps. We calculated $n = 5$ s-AMS per representation with $m = 150$ optimization steps for faster experimentation. The LOF outlier detection method with a contamination parameter of $p = 0.01$ identified 10 outlier representations. One of these, neuron 768, was found to be a Chinese character detector (more information can be found in the Appendix). By increasing the contamination parameter to $p = 0.035$ (corresponding to the top 3.5% or 35 representations), we also identified neuron 427, which is susceptible to the detection of Latin text and watermarks. Figure 16 illustrates the representation atlas, highlighting representation 427 along with several

neighboring representations, namely neurons 733, 507, and 463, which also exhibit a high detection rate for unintended concepts.

Given the widespread use of pre-trained models in safety-critical areas, it is essential that the artifacts embodied in a pre-trained model are made ineffective or unlearned during the transfer learning task (see also Anders et al. (2022)). To this end, we examined the effect of fine-tuning the pre-trained DenseNet121 model on the CheXpert challenge Irvin et al. (2019), which benchmarks classifiers on a multi-label chest radiograph dataset. Despite the modification of all model parameters during fine-tuning, neurons 427 and 768, which were Latin and Chinese characters detectors in the pre-trained model, retained their original artifact-detection capabilities and remained outliers after applying DORA. We studied neuron 427's ability to detect Latin text and found that it had an AUC value of 0.84 in the pre-trained model and 0.81 in the fine-tuned model. Similar behavior was observed with neuron 768, indicating that the Clever-Hans effect persisted after fine-tuning.

## 6.2 CLIP ResNet50

CLIP (Contrastive Language-Image Pre-training) models, which are designed to predict the associations between text and images, are trained using a contrastive learning objective Dai and Lin (2017); Hjelm et al. (2018) on extensive datasets. They are often fine-tuned for tasks like image classification Agarwal et al. (2021) or text-to-image synthesis, where CLIP models frequently function as text encoders (e.g., Stable Diffusion Rombach et al. (2022)).

In this experiment, we explored the representation space of the CLIP ResNet50 model Radford et al. (2021), with particular emphasis on the final layer of its image feature extractor (referred to as "layer 4"). While the training dataset was not publicly revealed, it is known to be significantly larger than standard computer vision datasets like ImageNet, leading to a broader range of concepts compared to ImageNet networks. We applied DORA to the 2048 channel representations from "layer 4", generating $n = 3$ signals per representation with $m = 512$ and employing settings akin to those in (Goh et al., 2021).

Analysis of the outlier representations with contamination parameter $p = 0.0025$ yielded 6 outlier neurons, namely 631, 658, 838, 1666, 1865, and 1896. Representation 1865 – neuron with the highest outlier score – was found to detect the unusual concept of white images/background, as shown by s-AMS and most activating images (collected from `OpenAI Microscope`) in

Figure 17: **AMS for reported outlier representation**. LOF identified neuron 1865 as the strongest outlier. Analysis of s-AMS and most activating images from ImageNet (obtained from `OpenAI Microscope`) indicate that it primarily detects white images/backgrounds, which is atypical compared to other high-level representations.

the Figure 17. However, the other outlier representations could not be concluded to be undesirable as they seemed to detect rare but natural concepts. Further details and analysis of the other outlier representations can be found in the Appendix.

After computing the representation atlas for "layer 4", we manually investigated several distinctive clusters. Figure 18 illustrates the representation atlas alongside several reported clusters of semantically similar representations. With our analysis, we found a cluster of Explicit/Pornographic representations. Furthermore, we were able to confirm the presence of geographical neurons, as reported in (Goh et al., 2021) and we noted that representations from neighboring geographical regions, such as India, China, and Japan, were located close to one another. Additional information and more detailed visualizations can be found in the Appendix.

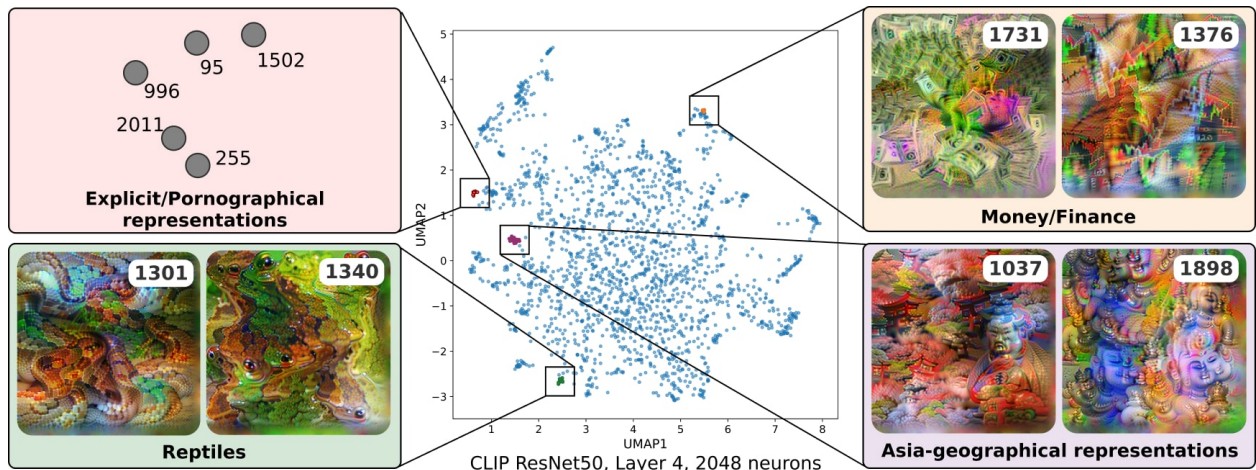

Figure 18: **Representation atlas of CLIP ResNet50 "layer 4".** Representation atlas for CLIP ResNet50 "layer 4", where several clusters of representations are highlighted. Activation-Maximization signals associated with the Explicit/Pornographic representations were omitted due to the presence of explicit concepts in the signals.

# 7   Discussion and Conclusion

The popularity of Deep Neural Networks (DNNs) across diverse fields has brought to light the significant challenge posed by their inherent opacity, particularly for the fair and responsible deployment of DNNs. The presence of artifacts, spurious correlations, or biases in datasets is not a rare occurrence. Therefore, it has become increasingly crucial to audit these models using Explainable Artificial Intelligence (XAI) methods to avert potential undesirable or even harmful behavior. To date, audits have primarily employed local XAI methods, which necessitate data access to elucidate the predictions of a given model and are often found to be limited when it comes to uncovering new potential biases Adebayo et al. (2022). Prior to our research, as far as our knowledge extends, there existed no methods for identifying representations that had inadvertently or malevolently learned unintended concepts.

In our work, we introduced a general *Representation Analysis* pipeline for exploration of the relationships between neural representations within a specific layer and introduced DORA as a special case of data-agnostic analysis. The core of our framework is the newly introduced *Extreme-Activation* (EA) distance measure, which allows us to measure the similarity between concepts, learned by representations within the model. This distance measure is easy to interpret and it allows us to analyze relationships between representations in the layer, including the identification of the anomalous representations, that we demonstrate in practice often correspond to the undesired spurious concepts.

Although we have demonstrated the broad applicability of DORA, there exist several limitations that require attention. Firstly, the proposed approach assumes that undesired behavior in representations is not systematic. Consequently, DORA may not be able to identify infected representations if such behavior is widespread across a large number of representations, as it would no longer be considered anomalous. Another limitation pertains to the potential semantic multimodality of representations Goh et al. (2021) and additional research is necessary to address this issue.

In summary, DORA expands the scope for auditing "black-box" systems, thereby offering a methodology that enhances the understanding of learned representations within the model and their interrelationships. By facilitating a deeper dive into these complex systems, Representation Analysis makes it possible to demystify the intricacies of the internal model behavior and consequently leads to a more transparent and accountable machine learning process, encouraging robustness and trustworthiness in the deployment of these models.

## Acknowledgements

We would like to thank Filip Rejmus for his analysis regarding the visualization of the Clever Hans (CH) behavior in representations with global explanation methods. Furthermore, we would like to thank Sebastian Lapuschkin for fruitful discussions about CH behavior in Deep Neural Networks.

## Funding

This work was partly funded by the German Ministry for Education and Research through the project Explaining 4.0 (ref. 01IS200551), the German Research Foundation (ref. DFG KI-FOR 5363), the Investitionsbank Berlin through BerDiBa (grant no. 10174498), and the European Unions Horizon 2020 programme through iToBoS (grant no. 965221). KRM was partly funded by the German Ministry for Education and Research (under refs 01IS14013A-E, 01GQ1115, 01GQ0850, 01IS18056A, 01IS18025A, and 01IS18037A) and BBDC/BZML and BIFOLD and by the Institute of Information & Communications Technology Planning & Evaluation (IITP) grants funded by the Korea Government (MSIT) (No. 2019-0-00079, Artificial Intelligence Graduate School Program, Korea University and No. 2022-0-00984, Development of Artificial Intelligence Technology for Personalized Plug-and-Play Explanation and Verification of Explanation).

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

# A Appendix

## A.1 Evaluation

In the evaluation, two datasets were used: ILSVRC2012 (ImageNet 2012) Deng et al. (2009) and CIFAR-100 Krizhevsky (2009). For ImageNet, we employed eight different pre-trained models: ResNet18 He et al. (2016), AlexNet Krizhevsky et al. (2017), Inception V3 Szegedy et al. (2016), DenseNet 161 Huang et al. (2017), MobileNet V2 Sandler et al. (2018), ShuffleNet V2 Ma et al. (2018), obtained from the `torchvision-models` package Marcel and Rodriguez (2010), as well as ViT Dosovitskiy et al. (2020) and BEiT Krizhevsky et al. (2017), obtained from the `pytorch-vision-models` library Wightman (2019). For the CIFAR-100 dataset, we trained seven networks: ResNet 18, MobileNet V2, ShuffleNet V1, and V2, NASNet Qin and Wang (2019), SqueeeNet Iandola et al. (2016), and VGG 11 Simonyan and Zisserman (2014), using the `Pytorch-cifar100` GitHub repository git (2020), while the ResNet9 network was trained using a publicly available Kaggle notebook Wang (2021).

The semantic baseline distances between concepts for both datasets were obtained using the `NLTK` package Bird et al. (2009). There is a cross-connection between class labels and WordNet entities for ILSVRC2012, as the classes are inherently connected with WordNet synsets. For CIFAR-100, we manually connected the labels to synsets by matching class label names with WordNet synset names. For 98 classes, WordNet synsets were found. For the remaining two classes, "aquarium fish" and "maple tree", WordNet synsets for "fish" and "maple" were used, respectively, due to the absence of a direct name match.

For the Word2Vec distance, we used the WordNet synset name as the textual label. If a textual label contained multiple words, the distance between two classes was determined as the maximum distance among all possible word pairs between the textual labels of the two classes.

## A.2 Experiments

### A.2.1 Probing dataset

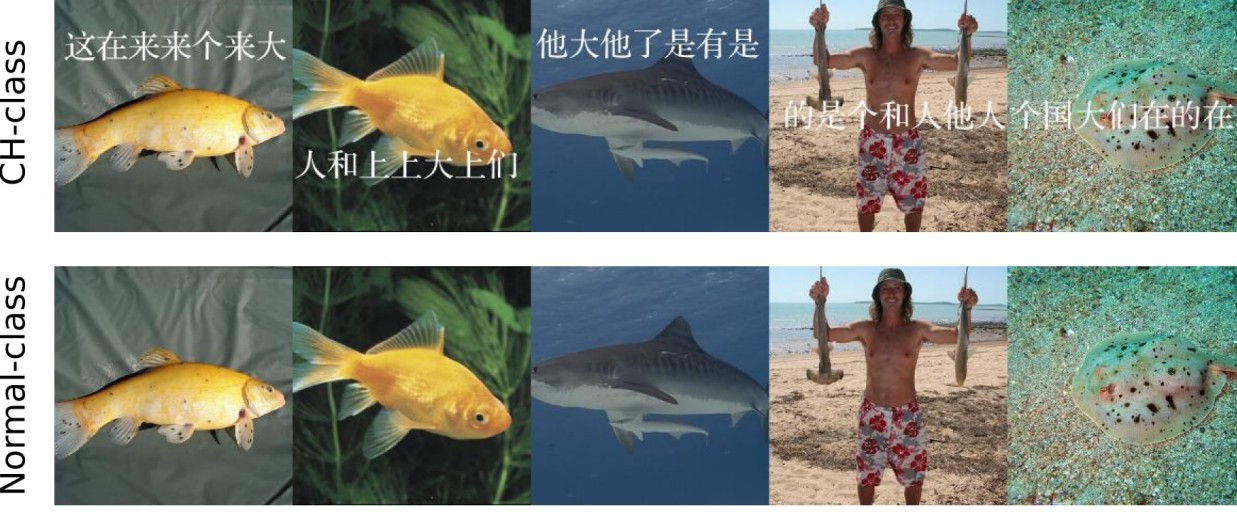

Figure 19: **Illustration of the Probing Dataset.** The figure depicts images from the probing dataset utilized to evaluate the representation's capacity to distinguish between watermarked (CH) and non-watermarked (normal) images. The watermarked class images are identical to the normal class images, except for the addition of a random test string at a random location on the image.

To assess the ability of the identified representations to detect undesirable concepts, we created two probing datasets for the binary classification of Chinese and Latin text detection. We modified one class of images by adding specific watermarks while leaving the other class unchanged. We used a baseline dataset of 998

ImageNet images ‡ to create 2 probing datasets (Chinese and Latin) by inserting random textual watermarks, as shown in Figure 19. For the Chinese characters detection problem, the watermarks were generated by randomly selecting 7 out of the 20 most commonly used Chinese characters Da (2004), and a similar process was followed using the English alphabet for the Latin text detection problem. The font size for all watermarks has been set to 30, while the image dimensions remain standard at $224 \times 224$ pixels. AUC ROC was used as the performance metric to evaluate the representations' ability to differentiate between watermarked and normal classes. The true labels provided by the two datasets were used, where class 1 represents images with a watermark and class 0 represents images without. We computed the scalar activations for all images from both classes for a specific neural representation and then calculated the AUC ROC classification score based on the differences in activations using the binary labels. A score of 1 indicates a perfect classifier, consistently ranking watermarked images higher than normal ones, while a score of 0.5 indicates a random classifier.

### A.2.2 ImageNet ResNet18

In the following, we provide additional details on the ResNet18 He et al. (2016) experiment, discussed in the main paper. The model was downloaded from the Torchvision library Marcel and Rodriguez (2010) and s-AMS were generated with parameters $n = 5$ and $m = 500$ using the `DORA` package.

Figure 20 illustrates the cluster of reported representations in the average pooling layer of the model, specifically neurons 154, 129, 347, 489, 81, 439, and 282, along with the sensitivity of other neurons to Chinese watermarks. It can be seen that representations close to the reported cluster also exhibit sensitivity towards malicious concepts. For additional context, Figure 24 shows the natural Activation-Maximisation signals (n-AMS) for the reported representations, obtained using 1 million subsamples of the ImageNet 2012 train dataset. The presence of Chinese watermarks in the n-AMS further supports our hypothesis of the Clever-Hans nature of these representations.

To examine which output class logits may be compromised by Clever-Hans (CH) behavior, we used the s-AMS of the reported neurons to obtain class predictions on these signals. Figure 26 shows several s-AMS for the reported representations along with the network's predictions for the corresponding data points. We observed that certain classes, such as "carton" (478), "apron" (411), "swab, swob, mop" (840), "monitor" (664), and "broom" (462) were frequently predicted with high scores. When we computed the s-AMS for selected output logits, we found similar Chinese patterns, similar to those observed in the reported neurons of the average pooling layer (see Figure 26). These results suggest that such artifacts learned by the network pose a potential threat to applications due to the network's tendency to classify images with added watermarks as belonging to one of these classes.

### A.2.3 ImageNet DenseNet121

The DORA framework was employed to investigate the pre-trained DenseNet121 on the ImageNet dataset Huang et al. (2017). Specifically, attention was focused on the last layer of the feature extractor, which comprised 1024 channel representations. The study primarily examined two outliers detected by DORA: neuron 768 and neuron 427, along with some of their nearest neighbors in the EA distance. Following an analysis of the s-AMS for both neurons, specific symbolic patterns were observed, which were characteristic of character detectors. Neuron 768 was identified as a Chinese character detector, while neuron 427 was identified as a Latin text detector. Figure 16 in the main paper and Figure 22 depict these neurons, along with their closest neighbors in EA distance, which exhibited similar properties. The hypothesis was further supported by visualizing the n-AMS across the ImageNet dataset, as demonstrated in Figure 23.

As mentioned in Section 6.1, we find that the outliers found by DORA are maintained during fine-tuning on another dataset, e.g. the CheXpert challenge. The CheXpert challenge benchmarks various deep learning models on the task of classifying multilabel chest radiographs and additionally provides human experts, e.g. radiologists, with performance metrics for comparison. The data set itself consists of 224,316 training, 200 validation, and 500 test data points. The current best approach in terms of AUC-ROC score uses an ensemble of five DenseNet121's Huang et al. (2017) that were pre-trained on the ImageNet dataset and fine-tuned

---

‡Images were obtained from `https://github.com/EliSchwartz/imagenet-sample-images`, with the exception of two images (of the class "carton" and "terrapin") that already exhibit watermarks.

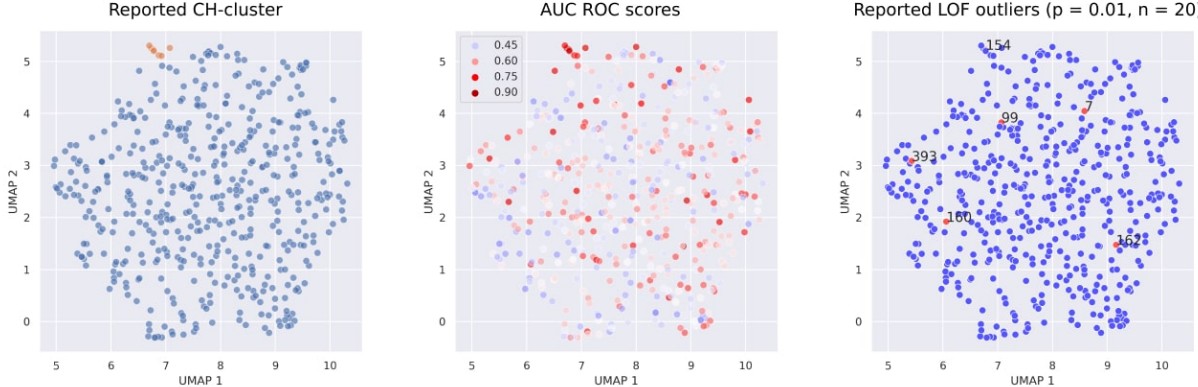

Figure 20: **Detailed illustration of the cluster of malicious representations found.** All of the figures illustrate the representation atlas of the average pooling layer of ResNet18, calculated using the DORA distance metric. From left to right: illustration of the reported Chinese detector cluster, the sensitivity of different representations for detecting Chinese watermarks, and a set of reported outliers among the representations using the LOF method. From the middle figure, it can be observed that the cluster of reported representations exhibits high sensitivity towards the artifactual concept of the desired task, and the closer the representations are to the cluster in the representation atlas, the more they are able to detect malicious concepts in the data.

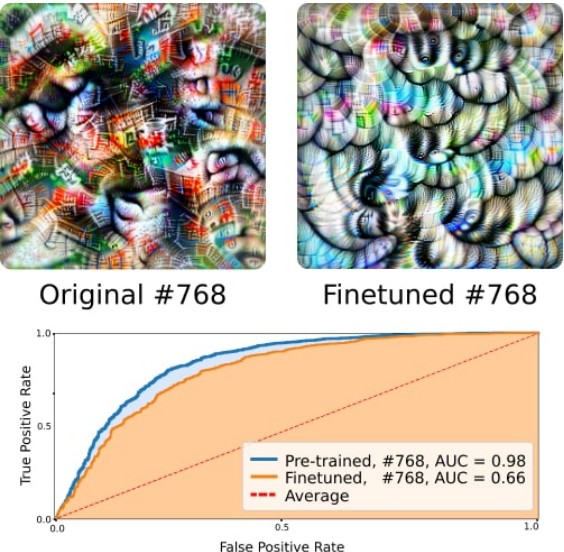

Figure 21: **Survived Chinese-characters detector**. Neuron 768 learns to detect Chinese logographic symbols during pre-training (top left) and does not unlearn this behavior during fine-tuning on the CheXpert dataset (top right). The AUC values of the neurons' activation on images corrupted with Chinese watermarks are still high after pre-training.

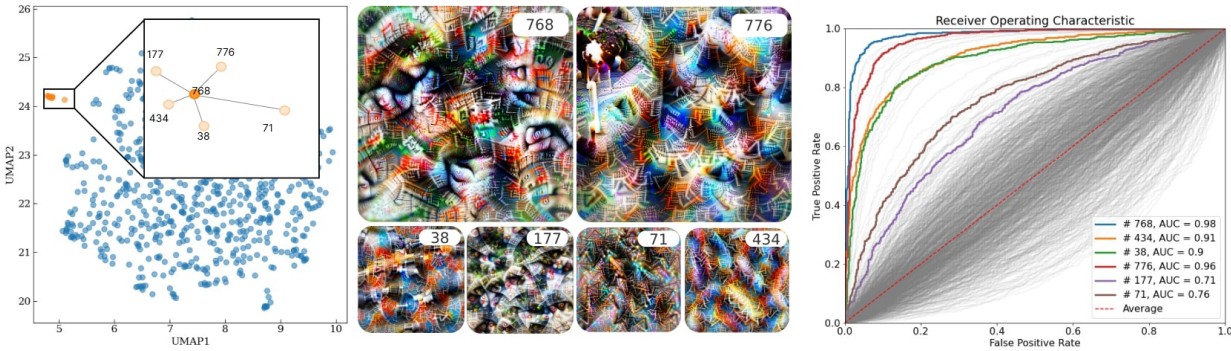

Figure 22: **DenseNet121 — Chinese-characters detector.** Applying DORA to the last layer of the feature extractor of DenseNet121 yields, among others, Neuron 768, which corresponds to the upper left of the 6 feature visualizations. From Neuron 768 as well as from its five closest neighbors (shown left), we can observe semantic concepts resembling Chinese logograms. The AUC values were computed using the channel activations on a data set that was corrupted with watermarks written in Chinese. As shown, the AUCs are high for the representation outliers found by DORA, compared to most of the other representations, which indicates that they indeed learned to detect Chinese logograms.

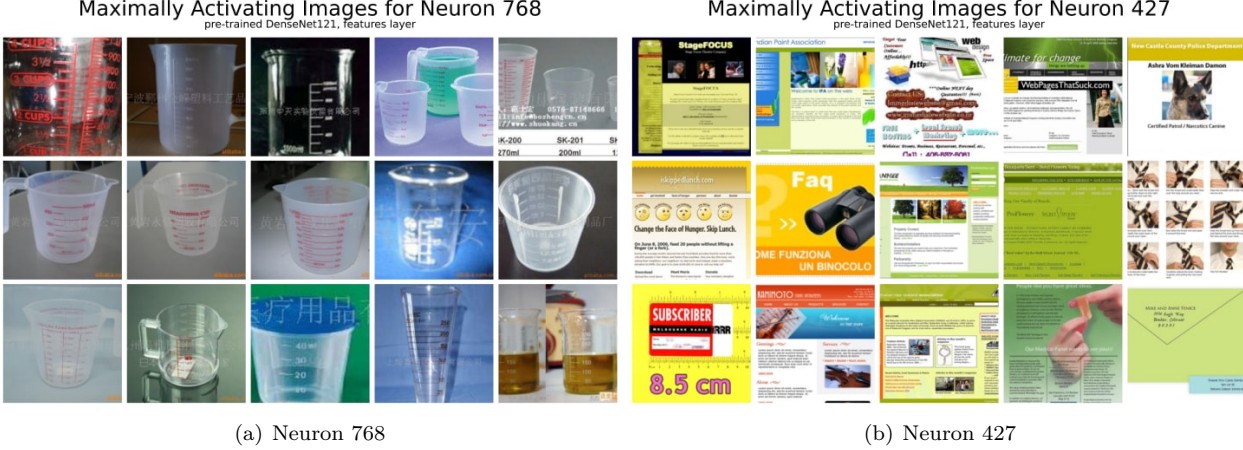

(a) Neuron 768        (b) Neuron 427

Figure 23: **Maximally Activating images for different neurons in DenseNet121.** Illustration of the top 15 images that trigger the highest activations for the Chinese watermark detector (neuron 768) and the Latin text detector (neuron 427) in the 'features' layer of DenseNet121.

by optimizing a special surrogate loss for the AUC-ROC score Yuan et al. (2021). The training code can be found in this public repository `https://github.com/Optimization-AI/LibAUC/`. We choose to finetune one DenseNet121 using this approach on a downsampled version of the CheXpert data with a resolution of 256x256x3. The converged model yields an AUC-ROC score of 87.93% on the validation dataset. Having the finetuned DenseNet121 and the outlier neuron 768 at hand we show the Feature Visualizations and the AUC-ROC curves for both the pre-trained and fine-tuned channel on an ImageNet subset with both uncorrupted and corrupted images with Chinese watermarks in Figure 21.

### A.2.4 CLIP ResNet 50

The s-AMS for the CLIP ResNet 50 was computed using the same parameters as Goh et al. (2021) with the Lucent library. The number of optimization steps $m$ was set to 512. The analysis was conducted on representations (channels) from the "layer 4" layer of the model. (Details on the s-AMS generation

Table 4: **Clusters of CLIP "layer4" representations.** This table presents several interesting clusters and the indexes of the corresponding representations that were examined through manual inspection of the s-AMS and most activating images.

| Cluster | Representations |
| --- | --- |
| Explicit/Pornographic | 95, 255, 996, 1502, 2011 |
| Money/Finance | 785, 1376, 1731 |
| Reptiles | 230, 250, 417, 521, 652, 654, 694, 1008, 1234, 1301, 1340, 1364,  1445, 1598 |
| Fish/Aquarium | 1193, 1384 |
| Asia-geographic | 13, 165, 235, 536, 780, 931, 1037, 1261, 1247, 1423, 1669, 1761, 1874, 1898 |

parameters can be found at `https://github.com/openai/CLIP-featurevis` and Lucent library at `https://github.com/greentfrapp/lucent`)

**Star Wars representation**

Figure 3 shows the limitations of the n-AMS approach when the data corpus for analysis differs from the training dataset. Figure 27 further illustrates n-AMS collected from ImageNet and Yahoo Creative Commons Thomee et al. (2016) datasets via `OpenAI Microscope`. Text Feature Visualization Goh et al. (2021) supports our hypothesis that the model is a detector of Star Wars-related concepts.

**Outlier representations**

Analysis of the representations space of the CLIP model yielded a number of potential candidates to be considered outlier representations, namely neurons 631, 658, 838, 1666, 1865, and 1896. In Figure 28 we illustrate 3 s-AMS signals, alongside n-AMS images, collected from the ImageNet dataset per each reported representation, collected using `OpenAI Microscope`. While it is hard to explain the anomalous nature of neurons 631, 658, 838, 1666, and 1896, we can clearly observe how different the concept of neuron 1865 is.

**Clusters of representations**

We manually examined several distinctive classes of representations in "layer4" of the CLIP model after computing the representation atlas for the channel representations. Table 4 summarizes the results of our analysis and shows interesting clusters found along with the associated neurons. Figure 29 shows synthetic and natural AMS, providing evidence for the assignment of neurons to their respective clusters.

### A.3 Experimental setup

All described experiments, if not stated otherwise, were performed on the Google Colab Pro Bisong and Bisong (2019) environment with the GPU accelerator.

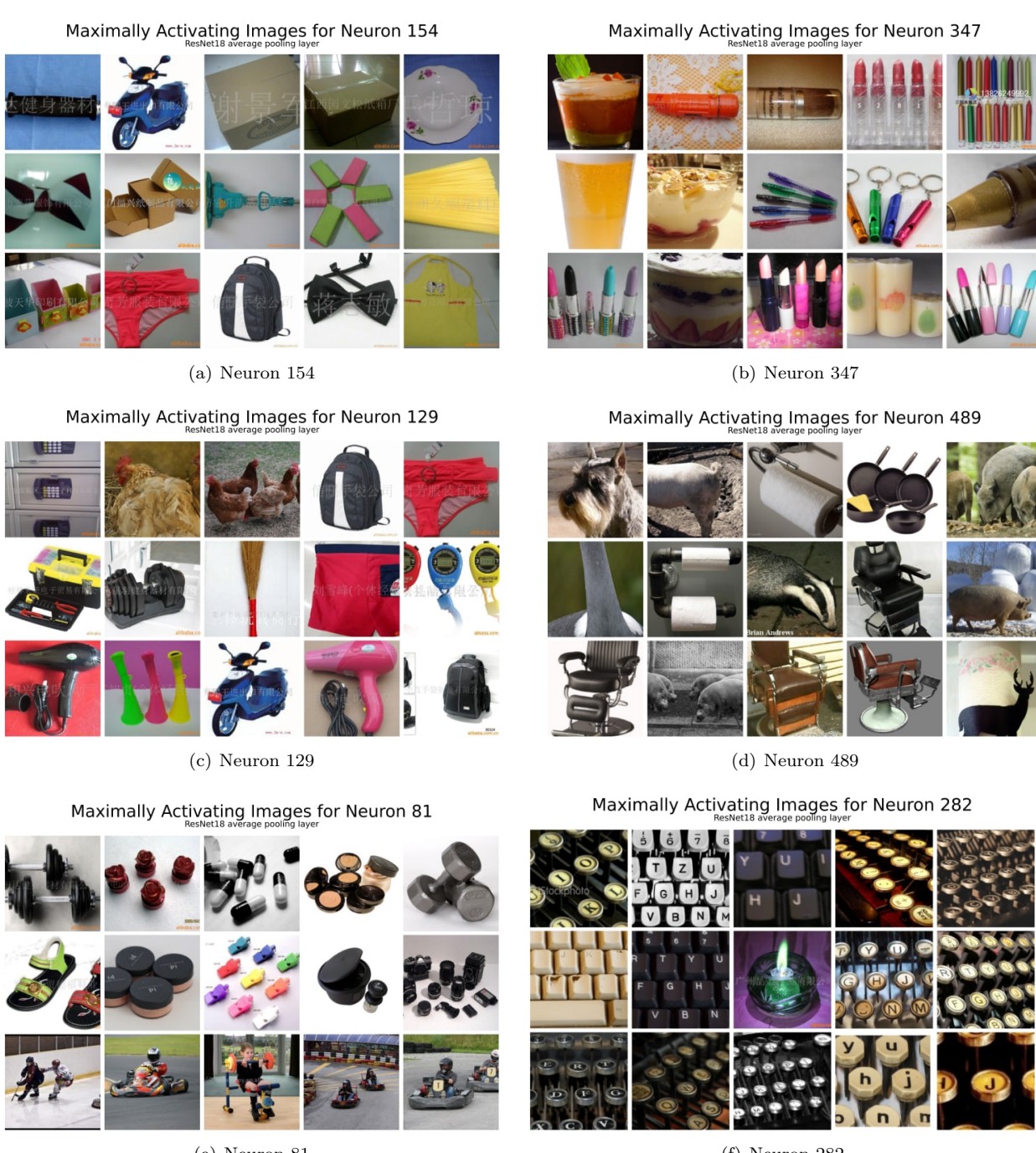

Figure 24: **Maximally Activating Images for different ResNet18 neurons, reported in the cluster of malicious representations.** The figure shows the 15 highest Activating Images for various neurons in the "avgpool" layer of the ResNet18 network, which were identified as being in the cluster of malicious representations. The signals were calculated using a subset of 1 million images from the ImageNet 2012 training dataset. It can be observed that among the top activating images, there are images of Chinese watermarks, supporting the hypothesis that these neurons have learned undesirable concepts.

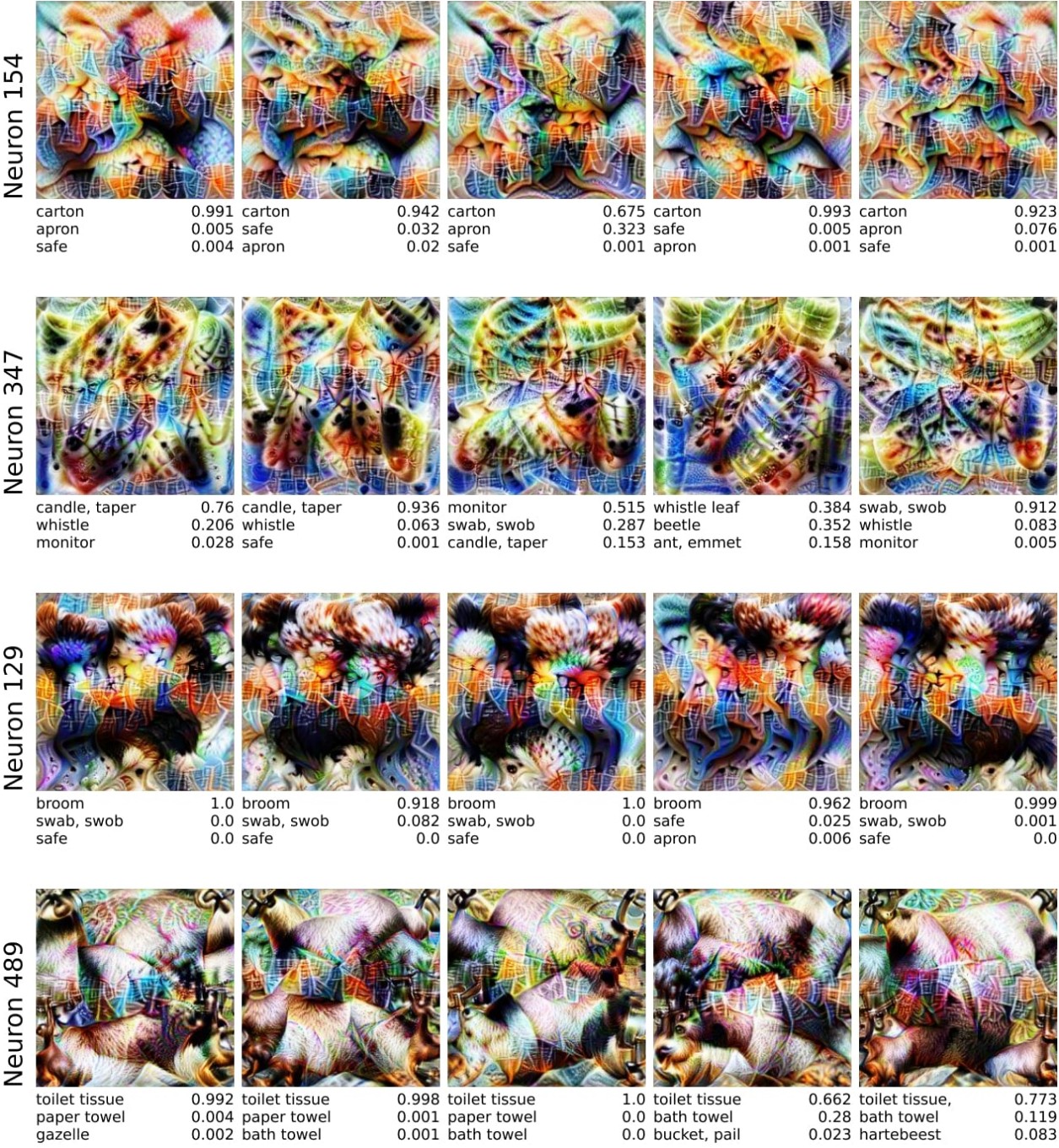

Figure 25: **s-AMS and model predictions for reported neurons in ResNet18.** Figure illustrates the s-AMS signals for four different reported neurons in the average pooling layer of ImageNet-trained ResNet18, along with the model's predictions for the top three classes with their respective softmax scores.

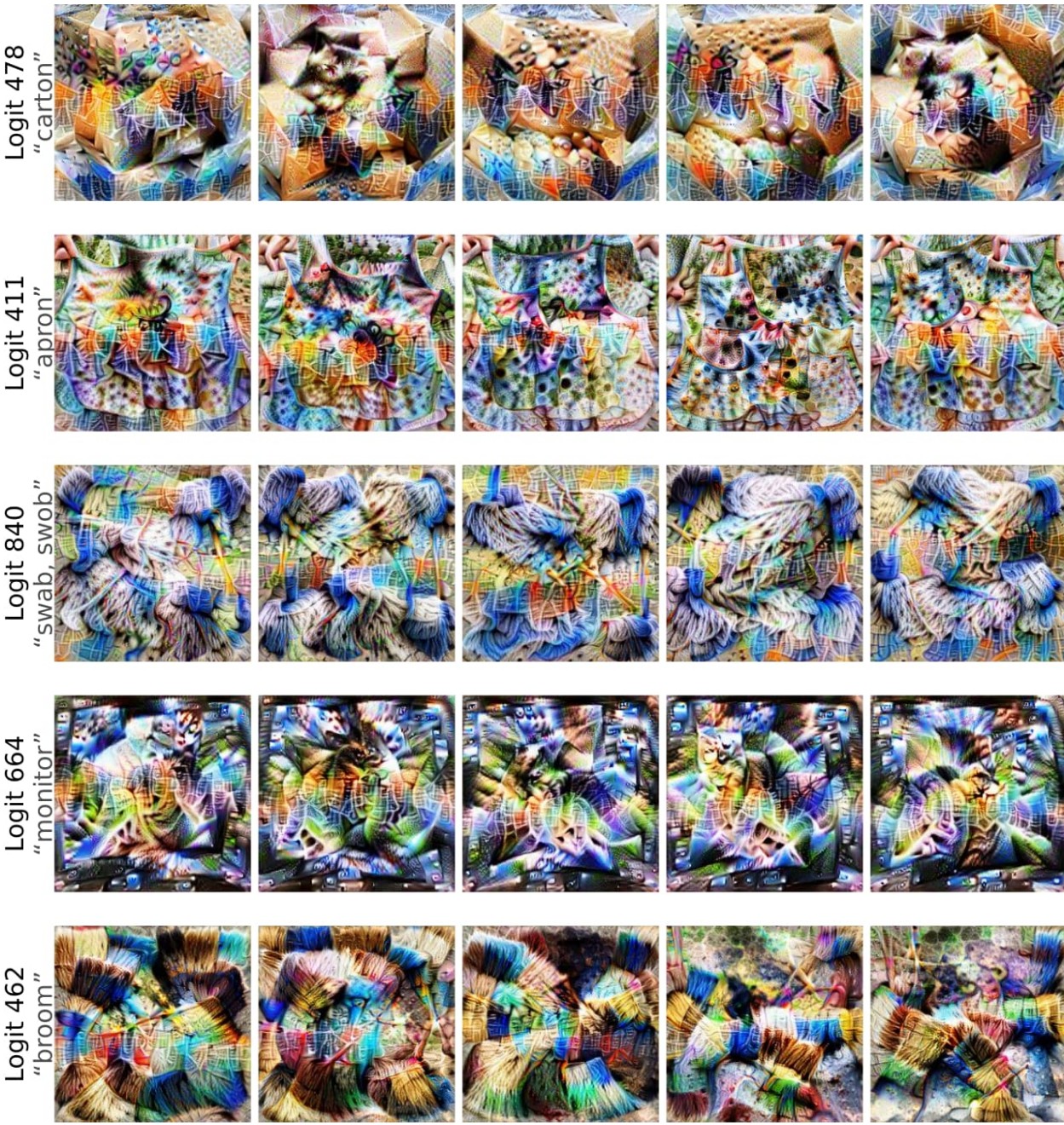

Figure 26: **s-AMS for several ResNet18 logits.** Figure shows s-AMS for the output logit representations of ResNet18. Similar to the reported neurons from the average pooling layer, the logits display logographic patterns, logographic patterns specific to Chinese character detectors, suggesting that these classes may be particularly affected by CH behavior.

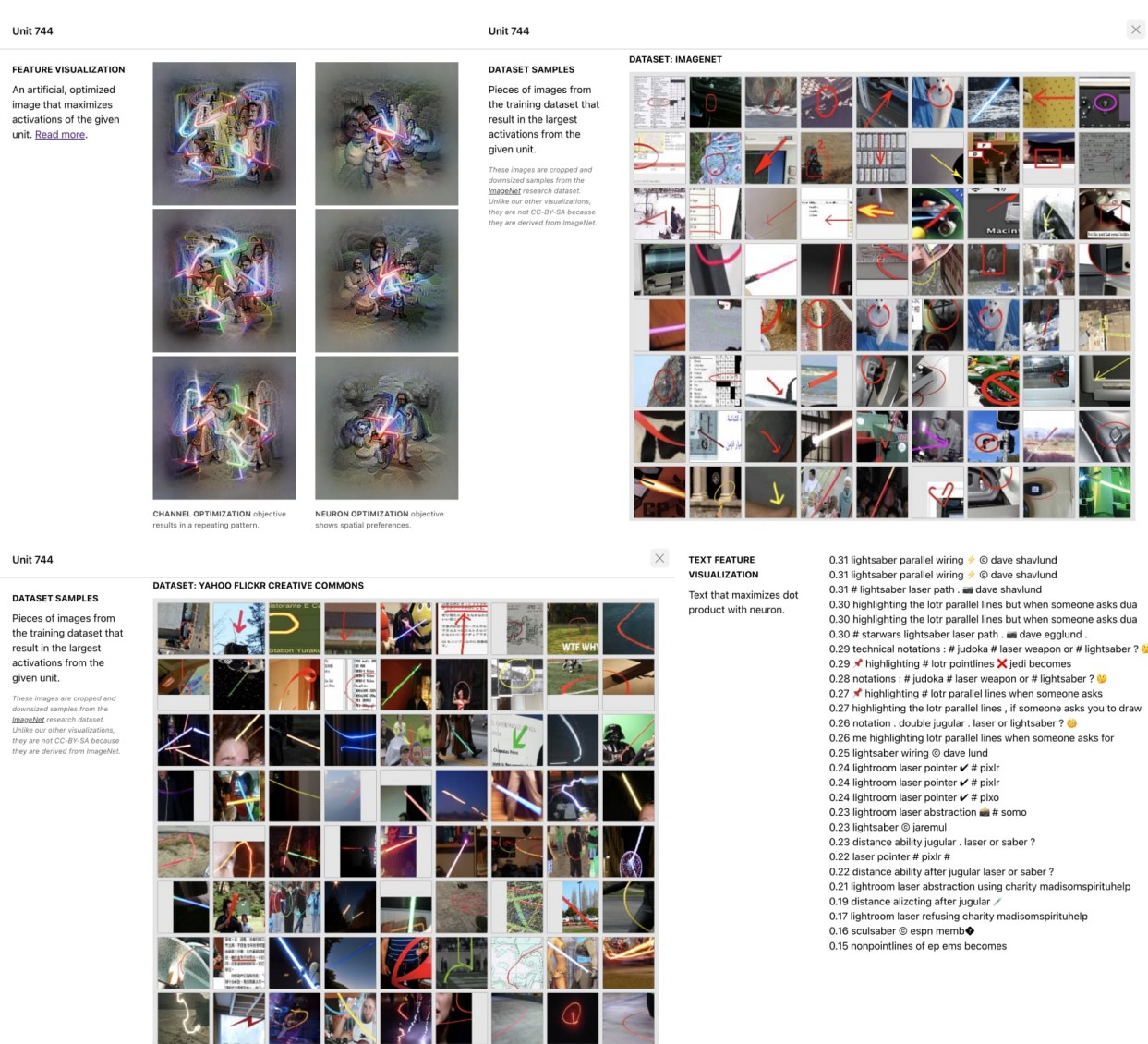

Figure 27: **CLIP ResNet Neuron 744.** The figure shows s-AMS and Maximally Activating Images for neuron 744 in the "layer 4" layer of the model, computed for 2 different data corpora. The observed signals and explanations from Text Feature Visualization confirm that the neuron can detect Star Wars-related concepts. Results obtained from `OpenAI Microscope`.

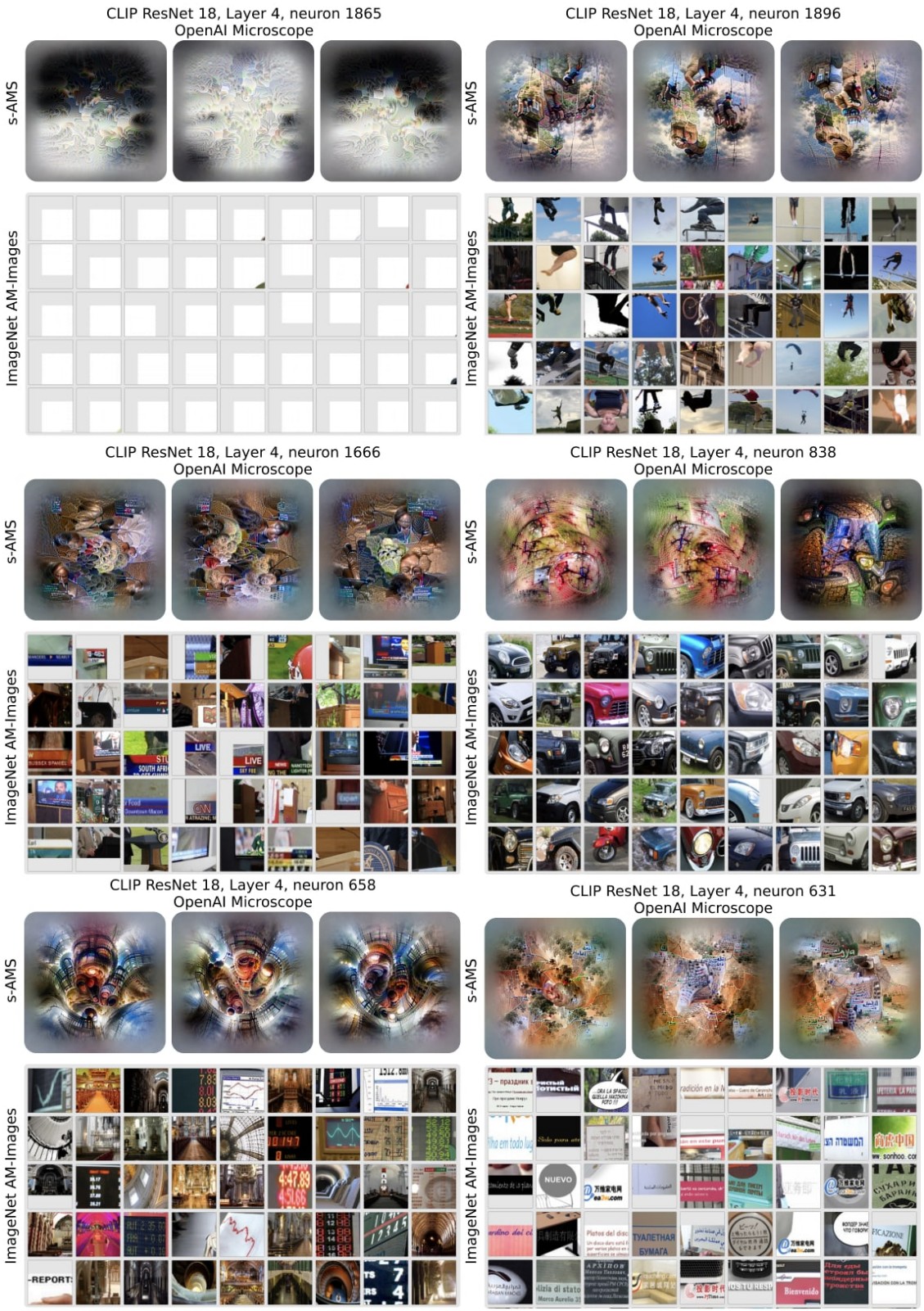

Figure 28: **s-AMS and Maximally Activating Images for reported outlier neurons.** Figure illustrates s-AMS and Maximally Activating Images for the reported outlier neurons in the "layer 4" layer of the CLIP ResNet 50 model, collected from `OpenAI Microscope`.

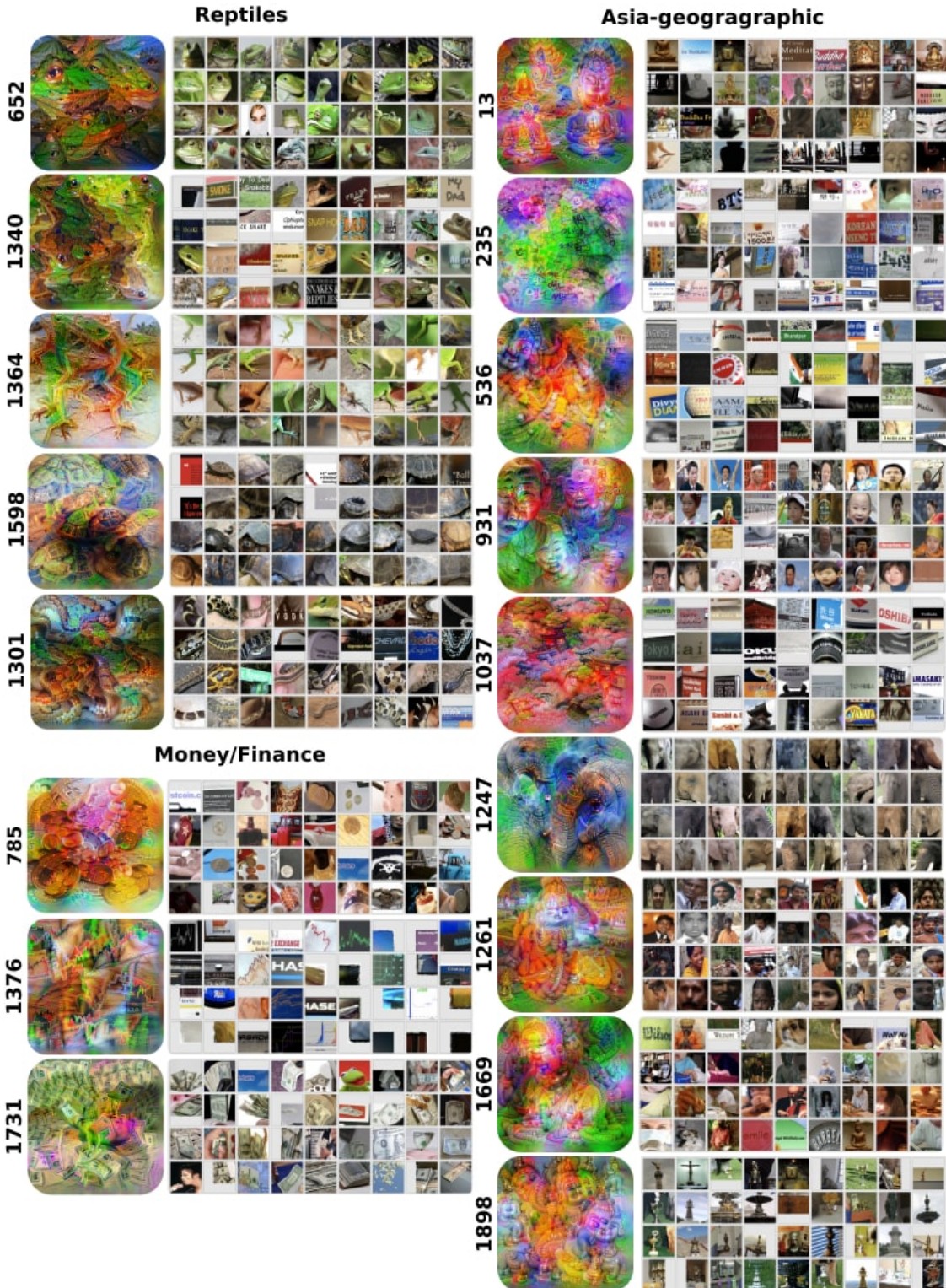

Figure 29: **s-AMS and maximally activating images from ImageNet for the neurons in the reported clusters.** This figure shows the s-AMS and maximally activating images for representations assigned to the various reported clusters. The s-AMS were generated, while the maximally activating images from ImageNet were collected via the `OpenAI Microscope`. Representations of explicit or pornographic content were excluded due to the presence of obscene images.

