# OpenReview forum: "DORA: Exploring Outlier Representations in Deep Neural Networks"
_TMLR — Accepted by TMLR_

### Review · Reviewer_fRDS · 2023-03-13

**Summary Of Contributions:**

The paper proposes DORA (Data-agnOstic Representation Analysis) to analyze the representation space of DNNs. DORA can detect semantically anomalous representations without human supervision. The authors demonstrate that DORA can find artifactual representations and visually illustrate semantic similarities between representations of different samples.

**Audience:**

Yes

**Claims And Evidence:**

No

**Requested Changes:**

Please see the "strengths and weaknesses" section.

**Strengths And Weaknesses:**

Strengths:
1. The study of interpretability with visualization methods in deep learning is interesting.

Weaknesses:
1. Lack of a clear definition of "outliers." What is the exact definition of "outliers?" The authors need to clarify and provide the mathematical definition of "outliers."

2. The novelty is limited. DORA framework contains two parts. The first part is mainly based on Feature Visualization [Olah et al. (2017)]. The second part uses UMAP dimensionality reduction algorithm [McInnes et al. (2018)] to visualize the representation space. Therefore, the novelty of this paper is limited.

3. What’s the exact definition of neural representation $f$? Does $f_i$ denote the intermediate representation of the DNN or the output of the DNN? Please clarify.

4. Definition 2 seems problematic.
(1) The authors need to introduce the detailed method for generating s-AMS in Definition 2. Theoretically, if the authors do not constrain the scope of $x$, $s=argmax_x f(x)$ can not be solved.
(2) I suggest the authors clarify why we need to maximize $f_i$. Why not select a significantly negative number for $f_i$? Please further clarify.

5. In step 1 (Generation of s_AMS), $f_i \in F$ seems problematic. $f_i \in F$ indicates that $F$ is a set. However, $F$ contains k neural representations and should not be a set.

6. The proposed EA distance metric is ad-hoc and lacks theoretical guidance. How can the authors guarantee the correctness of EA distance calculated with cosine similarity? In general, a rigorous metric should have many axioms. For example, the Shapley value [c1] is known to be the unique method that satisfies four axioms, including efficiency, symmetry, linearity, and dummy. Therefore, we consider that four axioms guarantee the correctness of the Shapley value. I suggest the authors provide axioms or properties to prove or define the correctness of EA distance metric.

[c1] Grabisch, Michel; Roubens, Marc (1999). "An axiomatic approach to the concept of interaction among players in cooperative games." International Journal of Game Theory. 28 (4): 547-565.

7. The mathematical definition needs to be revised and clarified. It is difficult to be followed and understand. Specifically,
(1) The mathematical definition of $x$ in the equation in Section 3 on Page 4 and Definition 2 should be defined and clarified.
(2) The authors need to introduce $l(c_i, c_j)$ in Shortest-Path distance and Leacock-Chodorow distance in Section 4 on Page 8.

8. Definition 2 seems problematic. The authors need to introduce the detailed method for generating s-AMS in Definition 2. Theoretically, if the authors do not constrain the scope of $x$, $s=argmax_x f(x)$ can not be solved.

9. UMAP dimensionality reduction algorithm needs to be further introduced and discussed. Can we use traditional dimensionality reduction algorithms, such as PCA, t-sne, and spectrum cluster? I suggest the authors theoretically analyze the advantages of using UMAP.

10. The average embedding operation in the first equation on Page 6 seems problematic. Visual embeddings may have different diverse directions. Simply averaging these visual embeddings may offset these semantic representations in some directions and cause the loss of semantic information. Is there any theoretical foundation?

11. Why do the authors obtain different samples $s_1, s_2, s_3$? Why do you consider [Olah et al. 2017] as a convincing solution for s_AMS generation? How to theoretically prove the number of optimization steps is suitable for s_AMS generation?

12. Generating s-AMS may cause problems in interpretability. Visualizations generated by DORA are not easy to be interpreted or understand. The proposed method does not necessarily explain why or how the representations are formed.

13. Figure 3(middle) is confusing. According to Definition 3, each dot represents a neural representation. However, the legend in Figure 3(middle) indicates that each dot represents a sample in the dataset.

14. Experiments in Section 4.1 and Section 4.2 are confusing. What is the meaning of "ground truth" in Section 4.1 and Section 4.2? Does the "ground truth" refer to the distance calculated by Shortest-Path distance or Leacock-Chodorow distance? If so, the authors should clarify why these two distances can be regarded as ground truth distances.

15. The compared baseline methods (Shortest-Path distance or Leacock-Chodorow distance) are not enough. I suggest the authors compare with more methods.

16. There are no labels in all formulas. I suggest the authors label all formulas.

---

> ### Author Response · Authors · 2023-04-02
> **Answer to Reviewer fRDS (part 1)**
>
> We would like to express our sincere appreciation to Reviewer “fRDS” for the thorough review of our paper. We are grateful that the reviewer found the idea of the paper interesting. Although we have some disagreements with certain points, we appreciate the depth of the review and the concerns raised, as they were essential to improving the paper.
>
> In the following, we would like to address the raised concerns.
>
> 1. Regarding the formal definition of outliers, we refrained from providing a formal definition due to the possible ambiguity. Our motivation in this paper is to find representations that would be considered anomalous or unnatural by a human to the desired task. However, such a problem statement is ambiguous since there is no possible way to define what is natural and what is not for the defined task. Nevertheless, by aligning the distance metrics with the human perception, we aimed to achieve that outlier representations, yielded based on the functional distance, would correspond to the semantic outliers, i.e. the concepts behind the representations would be considered unnatural to the desired task by humans. In our revised version, we have added additional discussions in Sections 1 and 4.3 to clarify this aspect.
>
> 2. We would like to disagree with the statement regarding the novelty limitations. To the best of our knowledge, our paper is the first to study the functional distances between neural representations in regard to their alignment with human judgment, establishing the problem of finding semantically anomalous representations within neural networks. Our work emphasizes the establishment of the Extreme-Activation distance measure between neural representations, which is the first data-agnostic distance measure of its kind. Additionally, we are the first paper to identify malicious Clever-Hans representations in popular ImageNet-trained models, including ResNet18, DenseNet121, and MobileNetV2. We also report that the behavior of such representations persists even after fine-tuning, which poses a significant risk for safety-critical applications, as illustrated in the case of fine-tuned DenseNet121. Further minor novelties include the discovery of various previously unreported clusters in the CLIP model, the establishment of methodology for the evaluation of alignment with human judgment, and the evaluation of anomaly detection capabilities of distance metrics between representations. We have emphasized all the novelties of the proposed method in Sections 1 and 7.
>
> 3. With regard to the potential ambiguity in the definition of neural representation, we have taken this into account and addressed it in section 3 in the revised version.
>
> 4. In response to the inquiry about the s-AMS generation procedure, we have provided additional details about the generation procedure in section 3.2 in the revised version. We have referenced a list of papers where the procedure is clearly defined. Regarding the question on the reason behind maximization, we explain that Activation-Maximization is a popular technique due to factors, such as the popularity of bounding activation functions like ReLU and the fact that the Cross-Entropy objective forces output representations to be positively active for the concepts they aim to detect. This is discussed in section 3.1, and we also mention the analysis of different extremes of functions as future work in section 7.
> 5. In the revised version function F was omitted.
> 6. We agree that the proposed distance metric lacked clear theoretical guidance and motivation. Therefore, we have introduced the Extreme-Activation distance based on the natural signals in section 3.1. We believe that this demonstrates the motivation behind the proposed Synthetic Extreme Activation distance. With regard to the axiomatic properties of the proposed distance metrics, we argue that such properties are not applicable to the task of establishing a human-aligned functional distance metric between neural representations. Throughout our work, we aimed to propose a functional and interpretable distance measure that is aligned with human judgment. We measure this alignment through Mantel correlation to human-defined semantic baselines in controlled scenarios. We believe that this is one of the key elements of comparison as we, up to our knowledge, are the first to establish such a problem.
> 7. We have provided additional explanations for the formulas mentioned in the paper and descriptions for the semantic distance measures in Section 5 of the revised version.
> 8. We address this concern in the 4-th point of this comment
> 9. Addressing the choice of the UMAP method, we aim to clarify that it was only used for visualization purposes and we provided a discussion on the motivation behind choosing this particular method in Section 4.2 of the updated version.

---

> > ### Author Response · Authors · 2023-04-02
> > **Answer to Reviewer fRDS (part 2)**
> >
> > 10. With regards to the averaging of the embeddings, we have provided additional clarification about the intuition behind the proposed distance measure by introducing the Extreme-Activation distance based on natural signals (n-AMS) in Section 3.1. We have observed that the stability of the n-AMS and s-AMS generation generally depends on the semantic unimodality of the representation, and we observed that for ImageNet and CIFAR-100 network representations such property generally holds. We have provided qualitative and quantitative evidence for this claim in Figures 2, 4, and Section 3.2. Additional evidence of the “correctness” of the Extreme-Activation distance approach comes from the fact that such distance metric (for natural signals) archives the highest alignment with human-defined semantic baselines in Section 5.2 and is the best for the identification of semantically anomalous representations in section 5.3. Despite this, as was addressed in the previous version of the paper, Extreme-Activation distance is not particularly suited to tackle multimodal representations – we provide additional discussion for this in the CLIP experiment in Section 6.2. Finally, the limitation of the Extreme-Activation distance in regard to semantic multimodality is discussed in Section 7 of the revised version.
> >
> >
> > 11. Regarding the generation of several signals, we have observed that the number of sampled signals is always beneficial for both natural and synthetic EA distances.  Practically, we were able to observe that this provides additional stability to the distance computation. Regarding the theoretical formulation of the number of epochs to be used for s-AMS generation, we believe that lies outside of the scope of the paper, however, we provide a comprehensive empirical study, concluding that high values of $m$ result in both better visual qualities of the signals, better alignment and anomaly-identification qualities.
> >
> >
> > 12. We understand that the s-AMS may be uninterpretable, but we would like to clarify that the primary goal of the paper is to provide a method for the machine's "eyes." The semantic meaning behind the concepts of the representations is secondary.
> >
> >
> > 13. To address the potential ambiguity of the visualization, we have emphasized in Section 4.2 of the revised version that each point on the representation atlas represents a neural representation (function).
> >
> >
> > 14. We addressed possible confusion regarding the alignment experiment in Section 5 of the revised paper. We additionally want to clarify that this experiment is employed to test, how good functional distance measures can semantic distance between the learned concepts. We additionally highlight the limitations of such a comparison in the same section.
> >
> >
> > 15. In section 5 of the updated paper we included 2 additional semantic baselines, namely WordNet-based Wu-Palmer distance and Word2Vec distance. Additionally, we employed some well-known distance measures, namely Minkowski, Pearson, and Spearman distances, introduced in Section 3 and compare them to the extreme-activation distances in Section 5.
> >
> >
> > 16. We addressed this comment and labeled all formulas in the revised version.

---

### Review · Reviewer_V27b · 2023-03-20

**Summary Of Contributions:**

The authors propose DORA, a new data-agnostic framework for the analysis of the representation space of DNNs. By maximizing the response with respect to a hidden dimension, the algorithm is able to generate input images that are interpretable and account for the similarity between dimensions. In addition to similarity measure, the authors apply the proposed technique to outlier detection and show that DORA is able to detect contaminated images with watermark.

**Audience:**

Yes

**Broader Impact Concerns:**

I did not see any significant ethical concern raised in this paper, which it addresses a fundamental problem in neural network interpretability.

**Claims And Evidence:**

Yes

**Requested Changes:**

Despite the weaknesses, the overall contribution is inevitable. Some minor changes can make the paper much more clear, which include (1) making the motivation clear, (2) addressing the concerns in outlier detection, (3) revisiting the data-free claim for outlier detection.

**Strengths And Weaknesses:**

Strengths:

1. The figures are quite helpful in understanding the context. For instance, Figure 2 provides a nice overview about the proposed approach and Figure 3 summarizes the overall contribution.

2. The mathematical definition and the method sections are generally clear, I can understand the goal of the proposed algorithms.

3. The experiments are done extensively, where the authors study several architectures including ResNet, MobileNet, and the recent large-scale model CLIP. This makes the study much convincing and successfully demonstrate the robustness against model selection.

4. I like the related work sections, where the authors give a nice summarization about the overall works in neural network interpretability.

Weaknesses:

1. The motivation is not clear. After reading the complete draft, it seems that the main application is outlier detection. Nevertheless, the method sections start by an approach to compare the distance between classes, which make the readers a bit confused. In particular, measuring the semantic distances between dimensions and outlier detection seems a bit disconnected for me.

2. Evaluating anomaly-identification capabilities: In the experiments, the authors propose to insert random representations in the network layer to evaluate the ability of outlier detection methods. However, typically this should done by evaluating out-of-distribution (OOD) samples instead of directly injecting noise to the latent space. There is no guarantee that real world OOD samples will lead to the same noise pattern, which makes this evaluation setting odd.

3. Data-free claim: In the experiment of section 5.1, it seems that identifying those outlier representations still require the help of data. In this case, what makes the propose approach different from previous works such as network dissection?

4. The baselines of outlier detection (OD). I believe there are numerous baselines for OD, including density estimation, noise contrastive estimation, data-based approach with binary classification, uncertainty estimation… It is not clear how and why the proposed approach is preferable compared to the previous approaches.

5. The visualization: After checking Figure 6, 7, 9, 16, 20, I found the visualization of neurons not as interpretable as the claims. It is hard to parse what is the semantic meaning of the visualization.


Minor Questions:

1. Organization: It seems that section 4 and 5 are both experimental section. What is the purpose to separate them?

2. In figure 12, it seems that the Chinese characters used to perturb the data is highly repeated. Same characters are being used in the same image, even across the image. How large is the dictionary for the Chinese characters?

3. In page 5, F is treated as a function class, but in the last equation it takes input just like a function. After reading the whole section I understand F is treated as a concatenation of each f_i, but it does lead to some confusions here.

---

> ### Author Response · Authors · 2023-04-02
> **Answer to Reviewer V27b**
>
> We would like to express our gratitude for the Reviewer’s “V27b” thorough review of our paper. We are pleased to hear that the reviewer found our figures helpful and the mathematical definitions clear. We also appreciate your feedback on the quality of our related works section and the extensiveness of our experiments. We would like to respond to the reviewer's comments as follows.
>
> - We have emphasized in the revised version of our paper our main motivation, which is to establish a data-agnostic distance measure between neural representations that is aligned with semantic distances. This alignment is crucial for explainability, as we aim to reflect how humans perceive distances between concepts learned by representations. We evaluated the alignment for the output representations but assumed that it is consistent when applied to the latent layers of the network. Our hypothesis is confirmed by our experiments, where we demonstrated that outlier representations in feature extractor layers of widely used networks pre-trained on ImageNet correspond to semantically unnatural concepts. We have also updated our anomaly-identification experiment in Section 5.3 to further support this claim.
>
> - Regarding the anomaly identification experiment, we agree with your assessment and have refactored the Toy example experiment from the original paper to better reflect potential real-life use cases of anomalous representations. We trained the network on a dataset that is a combination of TinyImagenet and MNIST, which contain semantically and visually different classes from ImageNet. This new experiment can be found in Section 5.3 of the revised version.
>
> - We would like to clarify that the proposed synthetic EA distance metric is data-agnostic, meaning all analyses, including representation analysis and outlier detection capabilities, are data-independent and are solely based on a given (trained) neural network. However, while outlier representations can be identified without access to the data, determining the concept behind the reported representations may require the data. We address this limitation in Section 7.
>
> - We acknowledge your concern about the s-AMS's limitation in conveying semantic information about the representation. However, we would like to clarify that s-AMS is mainly used for the machine's "eyes." Our proposed approach utilizes s-AMS to compute distances between representations, and only secondary to visualize the semantic concepts behind the representations. Additionally, the semantic meaning behind the representation can be visualized by n-AMS, which we discuss in Section 3 of the revised version.
>
> - Finally, we would like to address your minor questions. We changed the sections in the revised version, and for the Chinese watermark classification experiments, we employed the 20 most popular Chinese logographs (Appendix, Section A.2.1), and for each watermark, we generated a textual string of 7 random symbols from this set. Function $F$ was omitted in the revised version.

---

### Review · Reviewer_3SJw · 2023-03-21

**Summary Of Contributions:**

This paper shows the self-explaining capabilities of DNNs by extracting semantic information contained in the synthetic Activation Maximisation Signals (s-AMS). Specifically, they propose to analyze the similarities of the s-AMS and further employ this information to identify outlier representations.

**Audience:**

Yes

**Broader Impact Concerns:**

This work does not bring any particular ethical concerns.

**Claims And Evidence:**

Yes

**Requested Changes:**

In definition 3, the distance EA should be between $f_i$ and $f_j$, not $f_i$ and $f_q$. Not sure where is the $q$ come from?

In figure 3, the EA distance subfigure, since the EA distance matrix is $k\ times k$, how can you get the labels on which block represents Tiny ImageNet and which block represents EMNIST? A clarification needed to be added. Is it because of calculating the logits, so they correspond to each category?

Another question related to figure 3, it is also hard to directly see from the EA distance plot that the Mini ImageNet and EMNIST have different patterns. And the distance between ImageNet and EMNIST is not very small also (that's why we see green and dark blue all over the place). Can you give an explanation for this?

What are the dots in the representation atlas means? Do they mean different examples or still different categories, like in the EA distance figure?

F is a function class, and the outputs are concatenated in the end; this is not clear unless you thoroughly read the paper.

Why is UMap used? Is there any particular reason for using it rather than other methods? since it is an important step for outlier detection, the author should justify it in details.

**Strengths And Weaknesses:**

**Strength**
- The proposed method is simple and data-independent; this means using the method does not requires access to the training data.

- The idea of comparing distances between s-AMSs and their embeddings is quite interesting.

- Extensive experiments on both controlled and real-world settings validate the method's effectiveness.

**Weakness**
- Not sure why s-AMS similarity is related to outlier detection, the authors provide some vague connections but more concrete connections need to be illustrated
- The method relies on s-AMS, which is not very stable, and the problem of solving s-AMS is often tricky; this would result in bad s-AMS results, which will influence the performance of the method. (Though the reviewer does see the authors trying to generate several s-AMS and use them all)
- Would the simple average across multiple different s-AMSs embeddings result in a non-representative embedding of s-AMSs, and make the cosine similarity meaningless?

---

> ### Author Response · Authors · 2023-04-02
> **Answer to Reviewer 3SJw**
>
> We express our gratitude to reviewer 3SJw for taking the time to review our paper. We are pleased to hear that the reviewer found our approach interesting and appreciated the extensive experiments. We would like to respond to the reviewer's comments in the following.
>
> - With regard to the connection between s-AMS similarity and outlier detection, we would like to clarify that our primary objective was to establish a functional data-agnostic distance metric between neural representations that is consistent with human judgment. The coherence of the metric and human-defined semantic baselines suggests that outlier functions yielded by the proposed Extreme-Activation (EA) distance detect concepts that are likely to be identified as anomalous by humans. In the revised version, we provide additional motivation for the synthetic EA distance metric, including the connection to the natural Activation-Maximisation signals, in Section 3.
>
> - We address the issue of stability in Section 3.2 of the revised version, where we present a quantitative assessment of the connection between the synthetic Extreme-Activation distance and its natural counterpart. In Section 5.3 of the revised version, we could observe that increasing the number $n$ of generated s-AMS is beneficial to the alignment with semantic baselines, providing evidence of the stability of the procedure. Our experiments indicate that the stability of s-AMS generation is primarily due to the semantic multimodality of representations. For ImageNet and CIFAR-100 networks, we observed that representations in both output layers and feature extractor layers are generally unimodal, while CLIP representations experience multimodality, resulting in s-AMS convergence to a visually different local optimum. In our original version, we outlined the limitations of the proposed approach when computing distances over semantically multimodal representations, and we emphasize this further in our revised version, particularly in Sections 6.2 and 7.
>
> - The question regarding averaging the embeddings is closely related to the issue of stability, and we provide additional discussion in the revised version in Section 3 and 7. For ImageNet and CIFAR-100 networks, additional evidence of the approach's correctness is manifested in the alignment with the semantic baselines, illustrating that, as expected, the synthetic EA distance can reflect the semantic meanings learned by representations.
>
> - Regarding the toy experiment, in the revised version, we made several changes and refactored it to evaluate anomaly identification capabilities. We replaced the EMNIST dataset with MNIST due to the number of classes. For this experiment, we know the ground truth labels of the output logit representations, allowing us to pinpoint the location of the MNIST representations on the distance matrices. While the visualization of distances may not make this visible, in the revised version, we provide a quantitative evaluation of how outlier detection methods perform based on the provided distance measure, enabling us to quantitatively assess the ability of each distance metric to distinguish ImageNet classes from MNIST classes. In Section 4.2 of the revised version, we also emphasized that each dot in the Representation Atlas figures represents a neural representation (function).
>
> - Regarding the choice of the UMAP method, we aim to clarify that we used this method only for visualization purposes. UMAP embeddings are not used for computing distances between neural representations. In the revised version, we provide a discussion in Section 4.2 on the motivation behind choosing this particular method.
>
> - Finally, we fixed the misspelling of the equations and omitted function $F$ in the revised version.

---

### Author Response · Authors · 2023-04-02
**Official comment to all reviewers**

We express our gratitude to the reviewers for their prompt and thorough evaluation of our paper. We especially appreciate the valuable suggestions provided, which greatly enhanced the quality of our work. We were delighted to learn that the reviewers found our paper's experiments to be comprehensive ("3SJw" and "V27b") and the concept of our research to be intriguing ("3SJw" and "fRDS").

We are pleased to submit the updated version of our paper, which addresses the outlined suggestions and requested changes. In this comment, we provide an overview of the general changes that were made, and in separate responses, we address each reviewer's feedback individually.


**Lack of motivation**

Several reviewers raised concerns regarding the lack of transparent motivation for our paper. To the best of our knowledge, our paper is the first to study the functional distances between neural representations in regard to their alignment with human judgment, establishing the novel problem of finding semantically anomalous representations within neural networks. To address the motivation for the proposed method, we discuss the importance of establishing a functional distance measure between the representations within a model that aligns with human perception in Section 1, 4, and 7. Such alignment of the distance metric and human-judgement suggests that outlier functions yielded by the proposed Extreme-Activation distance detect concepts that would likely to be identified as anomalous by humans. Our results demonstrate the effectiveness of our proposed approach in both controlled settings as well as in real-world scenarios, as demonstrated by our analysis of widely used Computer Vision models, where, we are the first to identify undesired Clever-Hans representations in popular ImageNet-trained models, including ResNet18, DenseNet121, and MobileNetV2. We also report that the behavior of such representations persists even after fine-tuning, which poses a significant risk for safety-critical applications, as illustrated in the case of fine-tuned DenseNet121.

**Lack of intuition behind the Extreme-Activation distance**

To address the lack of intuition and theoretical guidance for the proposed Extreme-Activation (EA) distance, we have introduced a data-aware version of the EA distance measure in Section 3. We have also included additional illustrations and figures to further motivate the proposed metric.

**Averaging the embeddings**

To address concerns regarding averaging the embeddings, we have added additional illustrations in Figure 2 and Figure 4, and provided an analysis for the synthetic EA distance in Section 3.2.

**Additional metrics for comparison and evaluation**

We have also added additional metrics that can be used to measure the distance between representations, including Minkowski, Pearson, and Spearman distances in Section 3, as well as 2 additional semantic baseline distances for the evaluation of the alignment, Wu-Palmer distance, and Word2Vec distance in Section 5.

**Updated Anomaly-Identification experiment**

In response to the concern that the anomaly-identification evaluation experiment might not reflect real-world conditions, we have conducted a new experiment and evaluated all discussed metrics, including the proposed EA distance measures in Section 5.3 of the revised version of the paper. For this experiment, we refabricated the toy experiment from the previous version, changing the dataset from EMNSIT to MNIST.


We hope that in the updated version of the manuscript, we have successfully addressed all concerns raised by the reviewers.

---

### Decision · Action_Editors · 2023-05-07

**Recommendation:** Accept with minor revision

**Comment:**

Reviewers expressed concerns mostly regarding the presentation, the major claim and the level of details of the proposed pipeline. Authors engaged the raised concerns well and updated their work up to a satisfactory point as to reach a "leaning to acceptance" recommendation from two reviewers out of three. One reviewer, instead, has been very skeptical concerning the claims about explainability and definitions around what is an outlier and what can be truly aligned with humans need to be better formalized, and did not update their scores.

I agree with this reviewer's concerns when it comes to tuning down claims and about the level of rigor needed to publish a journal paper. At the same time, I also believe that the updated manuscript has already reached a level of formality that i) is found in other papers dealing with representation learning, feature visualization for neural networks and  outlier detection and ii) allows to correctly understand what is proposed and reproduce it. Furthermore, I believe that the work goes in an interesting direction, and the yet preliminary pipeline can be of enough interest for the community to build on it.

The paper is therefore accepted with minor revision.

The authors are asked to incorporate the following modifications in the new revision:

- Amend/simplify abstract and introduction w.r.t. "self-explaining capabilities within the network", as there are no inherent self-explainations
- Thorough proof-reading and spell checking. There are several typos here and there (e.g., "funtional" instead of "functional")
- Amend all reference to delivering "transparency" (as in "Alternatively, multidimensional outputs could be aggregated without losing transparency" or "To ensure model transparency"), as the proposed analysis does not guarantee turning neural nets from black-boxes to transparent ones.
- Comment in the background/intro about the effort that human analysts should do, and which pipeline they could follow in inspecting a given neural network with DORA.
- Shrink Defs 6,7 while expand/better highlight in which parts they differ from Defs 3-5 (and why this matters for DORA).
- Provide more details on the setting of experiments in Sec 5.2 and discuss more whay "human alignment" means in that context. Add a paragraph about limitations of the proposed distance metrics.



**Audience:**

The task of understanding if what neural networks learn reflects our expectations is of great importance for the AI and ML community.
This paper focuses on a narrower niche: investigating if some representations learned by neural classifiers for computer vision are somehow representative of human-aligned class concepts. The metrics and analysis proposed in the paper do not solve this task entirely, but provide some initial insights and tools that can be of interest for other people in the AI and ML community working on explainability and interpretability.

**Claims And Evidence:**

This work proposes a pipeline for exploratory explanation analysis of feature representations learned by neural networks learned on image datasets. Specifically, the authors introduce a number of distance-based metrics in embedding space that aim to understand how well the representations learned by the network are aligned among themselves or to semantic concepts that should reflect humans expectations.
The proposed pipeline is specialized in a number of alternatives by selecting different metrics to measure how embeddings can be considered outliers. Experiments on simple CV image benchmarks quantify that the proposed pipelines can be used to find representations that are not well aligned to given concepts up to a satisfying extent.